# Analysis of SUMO1-conjugation at synapses

**James A Daniel[1], Benjamin H Cooper[1], Jorma J Palvimo[2], Fu-Ping Zhang[3], Nils Brose[1], Marilyn Tirard[1]\***

[1]Max Planck Institute of Experimental Medicine, Molecular Neurobiology, Göttingen, Germany; [2]Institute of Biomedicine, University of Eastern Finland, Kuopio, Finland; [3]Institute of Biomedicine and Turku Center for Disease Modeling, University of Turku, Turku, Finland

**Abstract** SUMO1-conjugation of proteins at neuronal synapses is considered to be a major post-translational regulatory process in nerve cell and synapse function, but the published evidence for SUMO1-conjugation at synapses is contradictory. We employed multiple genetic mouse models for stringently controlled biochemical and immunostaining analyses of synaptic SUMO1-conjugation. By using a knock-in reporter mouse line expressing tagged SUMO1, we could not detect SUMO1-conjugation of seven previously proposed synaptic SUMO1-targets in the brain. Further, immunostaining of cultured neurons from wild-type and SUMO1 knock-out mice showed that anti-SUMO1 immunolabelling at synapses is non-specific. Our findings indicate that SUMO1-conjugation of synaptic proteins does not occur or is extremely rare and hence not detectable using current methodology. Based on our data, we discuss a set of experimental strategies and minimal consensus criteria for the validation of SUMOylation that can be applied to any SUMOylation substrate and SUMO isoform.

**\*For correspondence:** tirard@em.mpg.de

**Competing interests:** The authors declare that no competing interests exist.

## Introduction

The covalent attachment of small ubiquitin-like modifiers (SUMOs) to proteins is mediated by a conserved process known as SUMOylation. This eukaryote-specific post-translational protein modification is executed by an enzymatic cascade that is analogous to ubiquitination machinery and involves E1, E2, and E3 enzymes. In mammals, four genes encode four SUMO isoforms, SUMO1-4, of which only SUMO1, SUMO2, and SUMO3 have been extensively studied, because SUMO4 appears to have an unusual function. The amino acid sequences of SUMO2 and SUMO3 are 99% identical, while SUMO1 is only ~50% identical to SUMO2 and SUMO3. Based on their sequence similarity, their ability to form conjugated chains, and their unconjugated protein pool and dynamic conjugation, mammalian SUMOs are categorized into the SUMO1 and SUMO2/SUMO3 subfamilies, with SUMO2 and SUMO3 being indistinguishable by antibody detection (*Flotho and Melchior, 2013*; *Geiss-Friedlander and Melchior, 2007*; *Johnson, 2004*). The process of SUMO maturation and SUMO removal from substrates is carried out by SUMO1/sentrin-specific peptidases (SENPs), the DeSU-MOylating isopeptidases DeSI-1 and DeSI-2, and the ubiquitin-specific peptidase-like 1 protein USPL1 (*Drag and Salvesen, 2008*; *Schulz et al., 2012*; *Hickey et al., 2012*; *Shin et al., 2012*). Multiple cellular processes are controlled by SUMOylation and the consequent changes in substrate protein localisation, stability, interactions, or function - in particular transcriptional regulation and nucleocytoplasmic transport.

SUMOylation has proven difficult to study compared to some other post-translational modifications, primarily because of the transient nature of SUMOylation, the generally low levels of SUMOylated proteins, and the scarcity of reliable detection tools, such as antibodies. Various approaches

have been pursued to study SUMOylation in vivo and in vitro, which can be broadly categorized as either focused or unbiased. A focused approach constitutes a targeted analysis of a protein of particular interest, its possible SUMOylation, and functional consequences thereof. An unbiased approach entails the identification of SUMOylation substrates by systematic screening, for example,, via affinity-based purification of SUMOylated proteins from a sample of interest and proteomic identification of SUMOylation substrates, which typically yields a large set of candidate SUMOylation substrates (*Barysch et al., 2014*; *Becker et al., 2013*; *Eifler and Vertegaal, 2015*; *Impens et al., 2014*; *Matsuzaki et al., 2015*; *Tammsalu et al., 2014*; *Tirard et al., 2012*; *Yang et al., 2014*; *Hendriks et al., 2017*). In the past, unbiased screens for SUMO substrates covered a wide range of model systems - from monocellular organisms and cell lines to mammalian tissues. They involved a broad spectrum of experimental approaches - from the isolation of SUMOylated substrates after overexpression of affinity-tagged SUMOs to the affinity purification of endogenously SUMOylated substrates from wild-type (WT) material using highly specific anti-SUMO antibodies, or from knock-in (KI) mutant organisms using antibodies to affinity-tags of tagged SUMOs (*Becker et al., 2013*; *Impens et al., 2014*; *Tirard et al., 2012*; *Hendriks and Vertegaal, 2016*; *Lamoliatte et al., 2014*; *Yang and Paschen, 2015*; *Tatham et al., 2009*).

Regardless of whether targeted or unbiased approaches are used, a given protein of interest must be stringently validated as a SUMO substrate, ideally in vivo and without the overexpression of components of the SUMOylation machinery to avoid 'off target' artefacts. Given the generally rather low abundance of SUMOylated proteins, such a stringent validation process is often challenging. In this context, our approach has been to employ a $His_6$-HA-SUMO1 KI mouse line that allows analyses at near-endogenous levels of SUMO1 expression and conjugation. The use of $His_6$-HA-tagged SUMO1 in this model allows the precise localisation and stringent enrichment of genuine SUMO1 substrates, in vivo and in vitro (*Tirard et al., 2012*; *Tirard and Brose, 2016*).

SUMOylation has been recognized as an essential post-translational modification in health and disease (*Flotho and Melchior, 2013*; *Yang and Paschen, 2015*). In addition, SUMOylation in neurons has attracted a growing interest over the past decade (*Berndt et al., 2012*; *Gwizdek et al., 2013*; *Henley et al., 2014*; *Schorova and Martin, 2016*). In parallel to several other groups, our work has focused on protein SUMOylation in neurons and synapses, and the corresponding functional consequences. Our first tool of choice in this regard has been the $His_6$-HA-SUMO1 KI mouse line, and to our surprise, our unbiased biochemical and immunostaining analyses of brains of these $His_6$-HA-SUMO1 mice did not detect any SUMO1 conjugates at synapses (*Tirard et al., 2012*). Our results are in contrast to data from several target-focused studies, which led to the proposal that a range of synaptic proteins are SUMO1-conjugated. Consequently, there is a growing controversy regarding the still prevalent view that SUMO1-conjugation of synaptic proteins is a functionally relevant regulatory process in the context of synapse function.

In an attempt to resolve this controversy, we used the $His_6$-HA-SUMO1 KI mice and the SUMO1 knock-out (KO) model along with WT mice as controls, and engaged in a two-pronged approach. First, we used biochemical methods to assess the SUMO1-conjugation of seven previously reported synaptic SUMO1 substrates in vivo and in vitro - namely synapsin1a, gephyrin, GluK2, syntaxin1a, RIM1, mGluR7, and synaptotagmin1, which are representative components of presynaptic and postsynaptic compartments (*Matsuzaki et al., 2015*; *Girach et al., 2013*; *Craig et al., 2015*; *Tang et al., 2015*; *Ghosh et al., 2016*; *Choi et al., 2016*; *Martin et al., 2007*). Second, we searched for evidence of the presence of SUMO1 in synapses using brain sub-fractionation and immunolabelling of WT samples and of SUMO1 KO samples as a negative control. Overall, we did not obtain evidence of pre- or postsynaptic protein conjugation by SUMO1.

## Results

### The $His_6$-HA-SUMO1 knock-in mouse model

The $His_6$-HA-SUMO1 KI we developed previously has proven to be a reliable tool for the validation of known and the identification of novel SUMO1 substrates (*Tirard et al., 2012*; *Tirard and Brose, 2016*). Indeed, the SUMO1-conjugation patterns in brain samples as assessed by comparative SDS-PAGE and Western blotting analysis of SUMO1 and HA are essentially identical in WT and $His_6$-HA-SUMO1 KI brain, all known and well-validated SUMO1 substrates we tested so far could be reliably

verified using His$_6$-HA-SUMO1 KI samples, and several novel SUMO1 substrate candidates we initially identified by screening His$_6$-HA-SUMO1 KI samples, such as Zbtb20, could be validated with independent methods and/or were also found by others with independent methods and tools (*Becker et al., 2013*; *Impens et al., 2014*; *Tirard et al., 2012*; *Hendriks et al., 2015*). These findings indicate that the His$_6$-HA tag does not substantially or systematically change the substrate specificity of SUMO1. However, a caveat of the His$_6$-HA-SUMO1 KI mice that needs to be considered is the fact that the overall level of SUMO1-conjugation levels as assessed by SDS-PAGE and anti-SUMO1 Western blotting is reduced by ~20% in the brains of His$_6$-HA-SUMO1 KI mice as compared to brains of mice of a WT control line (*Tirard et al., 2012*). Similarly, we found that the level of anti-SUMO1 immunostaining of nuclei of hippocampal His$_6$-HA-SUMO1 KI neurons is reduced by ~35% as compared to roughly age-matched cells from a WT control line (*Figure 1—figure supplement 1*). Of note, though, the corresponding ~35% reduction represents an overestimation of the actual effect, because the WT control cells were older than the His$_6$-HA-SUMO1 KI cells (day in vitro 12 vs. 11) and SUMO1-conjugation levels increase substantially in WT brain between postnatal days 5 and 10 (*Tirard et al., 2012*) as well as in cultured neurons between days in vitro 11 and 12 (own unpublished observations). Based on these considerations, we conclude that SUMO1-conjugation in His$_6$-HA-SUMO1 KI neurons is reduced by ~20–30%. This effect is likely compensated by conjugation of SUMO2 or SUMO3 and somewhat reduces the sensitivity of the His$_6$-HA-SUMO1 KI as a reporter of SUMO1-conjugation and of the subcellular localisation of SUMO1.

## Synapsin1a, Gephyrin, GluK2, RIM1, Syntaxin1a, Synaptotagmin1, and mGluR7 are not SUMO1-conjugated in vivo

### Design of biochemical experiments

For biochemical analyses of SUMO1-conjugation at synapses, we focused on seven proteins that had previously been reported as SUMO1-conjugates and that are representative of presynaptic and postsynaptic compartments, that is, synapsin1a, RIM1, syntaxin1a, gephyrin, GluK2, synaptotagmin1 and mGluR7 (*Figures 1–5*). Using His$_6$-HA-SUMO1 KI mice and WT mice expressing untagged SUMO1 as controls, we investigated the SUMO1-conjugation status of each of the proteins in brain tissue by performing immunopurification (IP) assays in the presence of the de-SUMOylation inhibitor NEM (when not stated otherwise). As minimum evidence of SUMO1-conjugation, we looked firstly for specific enrichment of the protein following IP with an antibody against HA, which in the His$_6$-HA-SUMO1 KI model enriches all SUMO1-conjugated proteins (*Tirard et al., 2012*; *Tirard and Brose, 2016*). To confirm specificity of the IP, WT brain was used as a negative control in these experiments, and Western blot analyses were performed to confirm the success of the anti-HA IP (*Figure 1—figure supplement 2A*). Secondly, we looked for a shift in the apparent molecular weight of the relevant immunoisolated proteins. Although the theoretical molecular weight of SUMO is ~12 kDa, its conjugation to substrate proteins typically increases the apparent size of the substrate by ~20 kDa as assessed by SDS-PAGE. However, the molecular weight shift caused by SUMOylation is markedly influenced by the localisation of the SUMO acceptor lysine in the target protein, so that SUMOylation near protein termini generally causes larger size shifts (*Perdomo et al., 2005*). Further, the extent of the apparent size shift can vary between substrates and depends on the size and structure of the substrate and on the type of SDS-PAGE gel used. As the SUMO1-conjugated variants of many proteins are of low abundance, we performed the reverse IP in many cases (i.e. IP of the proposed substrate protein instead of HA-SUMO1), and thus enriched and assessed putative SUMO1-conjugate candidates using candidate-specific antibodies, along with anti-IgG IP as negative control, and His$_6$-HA-SUMO1 or WT brain tissue. Here, substrate validation was contingent upon the presence of a band of the SUMO1-conjugated protein with a ~ 20 kDa shift in molecular weight in the candidate-specific IP sample but not in the control sample, and anti-HA immunoreactivity of the shifted band in the IP from KI brain. In some cases, we performed additional experiments with HEK293 cells overexpressing the relevant proteins.

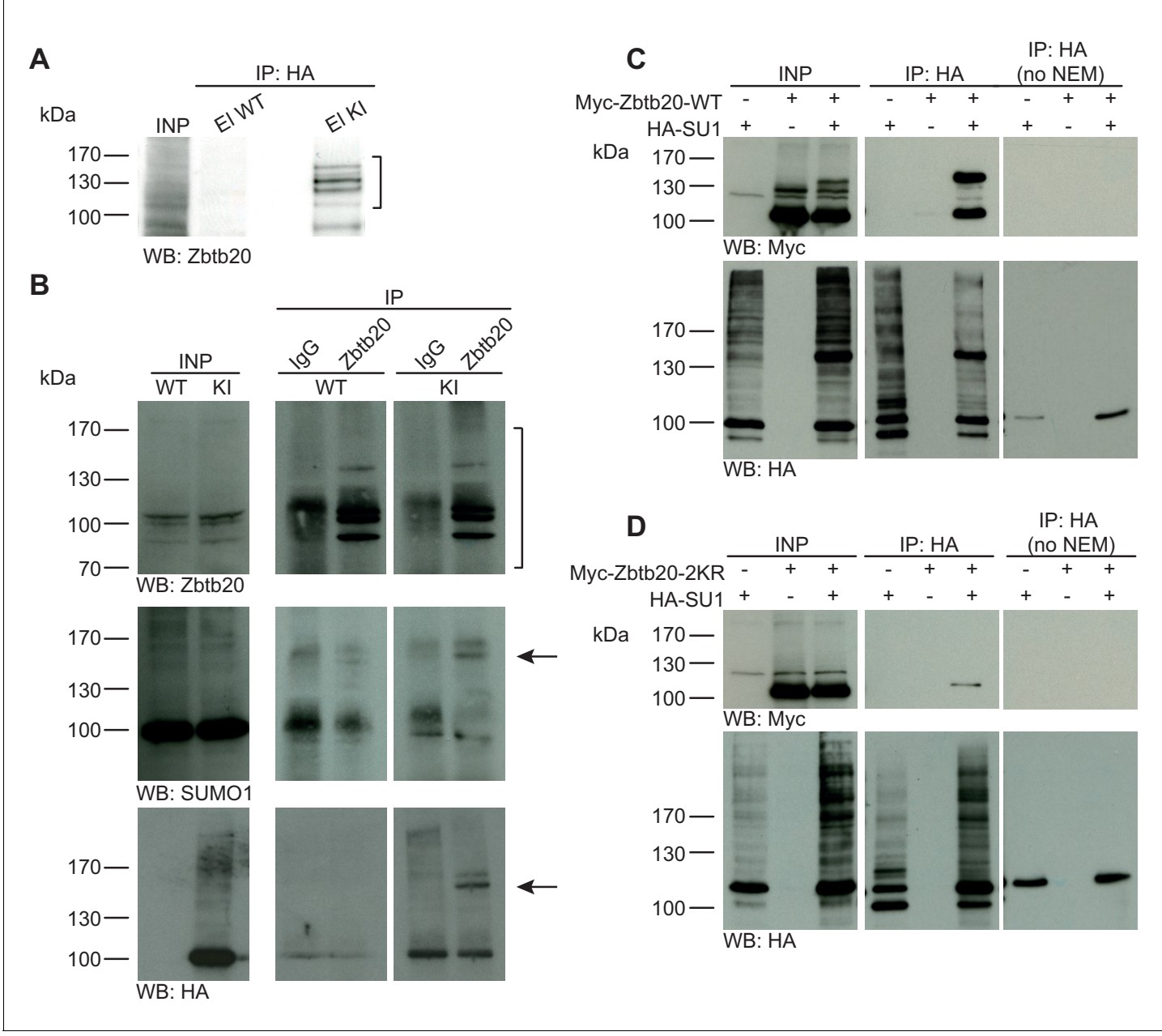

**Figure 1.** Zbtb20 is a SUMO1 substrate in vivo and in vitro. (**A**) SDS-PAGE (4–12%) followed by anti-Zbtb20 Western blot analysis of input and HA peptide eluate fractions from anti-HA immunoprecipitation in the presence of 20 mM NEM from WT and His$_6$-HA-SUMO1 KI brains. Zbtb20 is specifically enriched in KI eluate (bracket), indicating that Zbtb20 is a SUMO1 target in vivo. (**B**) Representative SDS-PAGE (8%) followed by anti-Zbtb20, SUMO1, and HA Western blot analysis of input and eluate fractions of anti-Zbtb20 and anti-IgG immunopurifications from WT and His$_6$-HA-SUMO1 KI brains in the presence of 20 mM NEM. Anti-Zbtb20 Western blot confirm the enrichment of Zbtb20 in both WT and KI brain samples solely when anti-Zbtb20 antibody is used (upper panel, brackets). Anti-SUMO1 Western blot analysis revealed a shifted band corresponding to SUMO1-Zbtb20 in both WT and KI (middle panel, black arrow). Anti-HA Western blot revealed a shifted band corresponding to His$_6$-HA-SUMO1-Zbtb20 solely in KI eluate (lower panel, black arrow). (**C**) SDS-PAGE (10%) followed by Western blot analysis of input and eluate fractions of anti-HA immunoprecipitation in the presence or absence of 20 mM NEM from HEK cells overexpressing HA-SUMO1 and Myc-Zbtb20-WT, alone or in combination. Anti-HA Western blot confirms enrichment of HA-SUMO1 conjugates in the presence of NEM while all SUMO1 conjugates (with the exception of RanGAP1) are lost when NEM is omitted (lower panel). Anti-Myc Western blot analysis confirms the co-enrichment of Myc-Zbtb20-WT in the presence of NEM and when co-expressed with HA-SUMO1. (**D**) SDS-PAGE (10%) followed by Western blot analysis of input and eluate fractions of anti-HA immunoprecipitation in the presence or absence of 20 mM NEM from HEK cells overexpressing HA-SUMO1 and Myc-Zbtb20-2KR, alone or in combination. Anti-HA Western blot confirms enrichment of HA-SUMO1 conjugates in the presence of NEM while all SUMO1 conjugates (with the exception of RanGAP1) are lost when

*Figure 1 continued on next page*

*Figure 1 continued*

NEM is omitted (lower panel). Anti-Myc Western blot analysis shows that Myc-Zbtb20-2KR is not co-enriched in the presence of NEM and when co-expressed with HA-SUMO1 (upper panel). Images are representatives of at least three independent experiments.

The following figure supplements are available for figure 1:

**Figure supplement 1.** SUMO1 levels in the nuclei of His$_6$-HA-SUMO1 KI neurons (also relates to *Figures 2–8*, which include results obtained with the His$_6$-HA-SUMO1 KI).

**Figure supplement 2.** Characterisation of the anti-HA immunoprecipitation and of the anti-Zbtb20 antibody.

## Zbtb20 as an example of reliable validation of a SUMO1 substrate in vivo and in vitro using the His$_6$-HA-SUMO1 KI

In a previous unbiased screen for genuine SUMO1 targets using the His$_6$-HA-SUMO1 KI, we identified many well-known SUMO1 substrates (*Tirard et al., 2012*) as well as several novel ones. The latter included the transcription factor Zbtb20, which was in parallel identified as a SUMO1 target in multiple independent proteomic screens (*Becker et al., 2013*; *Impens et al., 2014*; *Tirard et al., 2012*; *Hendriks et al., 2015*). To further validate the experimental design described above, we decided to analyse the SUMO1 conjugation status of Zbtb20 in vivo and in vitro. First, we evaluated the specificity of our anti-Zbtb20 antibody and performed a Western blot analysis of WT, heterozygous Zbtb20 KO, and homozygous Zbtb20 KO mouse brain homogenates (*Figure 1—figure supplement 2B*). Our anti-Zbtb20 antibody specifically recognized at least four specific bands, which were absent in Zbtb20 KO samples and had apparent molecular weights of ~100–130 kDa, matching known Zbtb20 variants. Having verified the specificity of our anti-Zbtb20 antibody, we next evaluated the SUMO1-conjugation status of Zbtb20 in vivo by performing IPs (*Figure 1A and B*). Analysis of anti-HA IP eluate material from His$_6$-HA-SUMO1 KI mice by SDS-PAGE (4–12%) and Western blotting using anti-Zbtb20 antibodies revealed a strong enrichment of size-shifted bands of Zbtb20 running between 100 and 170 kDa, indicating SUMO1-conjugation of Zbtb20 in vivo (*Figure 1A*), but no corresponding signal in the eluate fraction from WT mice. The presence of multiple Zbtb20 bands can be explained by the existence of multiple splice variants (*Figure 1—figure supplement 2B*) and the fact that Zbtb20 can oligomerise, so that non-SUMO1-conjugated Zbtb20 species are partly co-purified. Next, we conducted reverse IP experiments and enriched Zbtb20 using our previously validated anti-Zbtb20 antibody (*Figure 1—figure supplement 2B*) from WT and His$_6$-HA-SUMO1 KI brains, using anti-IgG IP as negative control (*Figure 1B*). SDS-PAGE (8%) and Western blot analysis of input and eluate fractions using anti-Zbtb20 antibodies revealed a strong and specific enrichment of various Zbtb20 isoforms in the test sample as compared to the anti-IgG control (*Figure 1B*, upper panel, brackets). Anti-SUMO1 Western blot analysis of the same fractions revealed the presence of a size-shifted band of Zbtb20 at ~150 kDa. This band likely corresponds to the largest Zbtb20 isoform (normally at ~130 kDa) being SUMO1-conjugated (*Figure 1B*, middle panel, black arrow), or to a smaller variant of Zbtb20 being modified by two SUMO1 moieties. Subsequent anti-HA Western blot analysis showed that this shifted band is only observed with the anti-Zbtb20 IP performed from His$_6$-HA-SUMO1 KI mice, but not from WT mice (*Figure 1B*, lower panel, black arrow). Altogether, these data confirm the SUMO1-conjugation of Zbtb20 in vivo.

Next, using an overexpression approach in HEK293 cells, we compared the SUMOylation status of WT Zbtb20 (Zbtb20-WT) and a mutant form, in which two candidate lysine residues (K330 and K371) are mutated to arginines (Zbtb20-2KR) (*Figure 1C and D*). Lysates of HEK293 cells overexpressing HA-SUMO1 and Myc-Zbtb20-WT (alone or in combination) were subjected to anti-HA IP in the presence of the de-SUMOylation inhibitor NEM. SDS-PAGE (10%) and Western blot analysis of input and eluate fractions showed that Myc-Zbtb20-WT, which normally appears between 100 and 130 kDa, was enriched in the HA eluate fraction only when NEM is present in the buffer to block de-SUMOylation as evident by the enrichment of a shifted band above 130 kDa (*Figure 1C*) - the non-SUMOylated form of Zbtb20 is likely co-purified due to oligomerisation between SUMO1-conjugated and non-conjugated forms of Zbtb20. Anti-HA Western blotting confirmed the efficient enrichment of HA-SUMO1 targets, especially when NEM was added to the lysis buffer

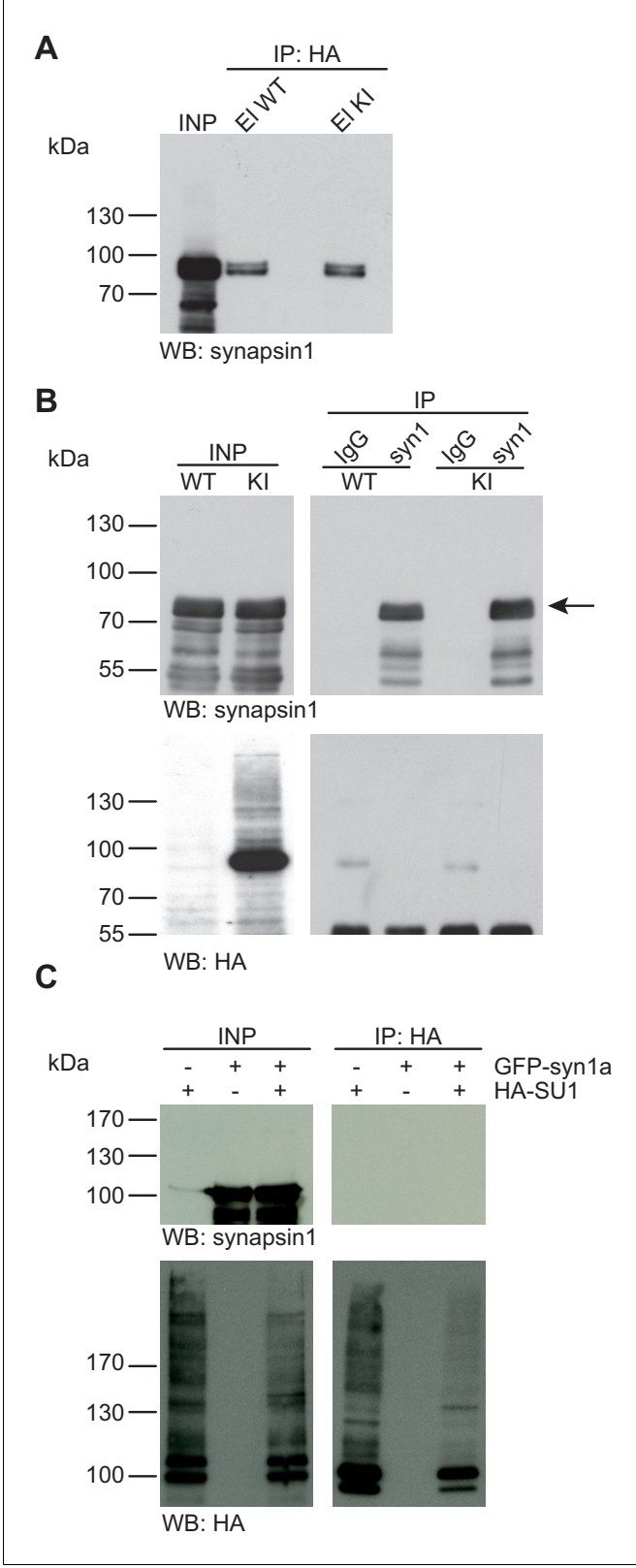

**Figure 2.** Synapsin1a is not SUMO1-conjugated in vivo and in vitro. (**A**) SDS-PAGE (4–12%) followed by Western blot analysis using anti-synapsin1 antibody of input and HA peptide eluate fractions from anti-HA immunoprecipitation in the presence of 20 mM NEM from WT and His$_6$-HA-SUMO1 KI brains. The presence of synapsin1 in both WT and KI eluates indicates non-specific binding of synapsin1a to the affinity matrix. (**B**)
*Figure 2 continued on next page*

*Figure 2 continued*

Representative SDS-PAGE (10%) followed by Western blot analysis of input and eluate fractions of anti-synapsin1 and anti-IgG immunopurifications in the presence of 20 mM NEM from WT and His$_6$-HA-SUMO1 KI brains. Anti-synapsin1 Western blot confirms enrichment of synapsin1 after affinity purification using synapsin1 antibody and not using nonrelated IgG (black arrow). However, anti-HA Western blot does not detect any SUMO1-synapsin1 shifted band. (C) SDS-PAGE (10%) followed by Western blot analysis of input and eluate fractions of anti-HA immunoprecipitation in the presence of 20 mM NEM from HEK cells overexpressing HA-SUMO1 and GFP-synapsin1a, alone or in combination. While anti-HA Western blot confirms enrichment of HA-SUMO1 species (lower panel), synapsin1a is not detected in the eluate samples (upper panel). Images are representatives of at least three independent experiments.

(*Figure 1C*, lower panel). On the contrary, Myc-Zbtb20-2KR was not enriched by anti-HA IP (*Figure 1D*) irrespective of whether NEM was present, indicating that Zbtb20-2KR is no longer SUMO1-conjugated (*Figure 1D*). Overall, our data on Zbtb20 show that, in principle, our His$_6$-HA-SUMO1 KI model, approach, and experimental design represent a powerful combination of tools and methods for the stringent biochemical identification and validation of SUMO1 substrates in vivo and in vitro.

## No evidence for SUMO1-conjugation of Synapsin1a in vivo and in vitro

Synapsin1a is a presynaptic vesicle-associated phosphoprotein that is thought to play an important role in synaptic vesicle organisation and mobilisation (*Song and Augustine, 2015*). A recent report indicated that synapsin1a is a SUMO1 substrate in vivo and in vitro, and that SUMO1-conjugation of synapsin1a affects the availability of synaptic vesicles for transmitter release (*Tang et al., 2015*). First, we examined SUMO1-conjugation of synapsin1a using the His$_6$-HA-SUMO1 KI mouse model (*Figure 2A*). We performed anti-HA affinity purification of proteins from WT and His$_6$-HA-SUMO1 KI brain lysates, followed by SDS-PAGE (4–12%) and Western blot analysis of input fractions and eluates from the IP matrix. Western blot analysis using an anti-synapsin1 antibody (an antibody that recognizes both synapsin1a and synapsin1b) revealed a signal at the expected molecular weight of synapsin1 in eluates from both, WT and His$_6$-HA-SUMO1 KI samples, indicating that synapsin1 binds non-specifically to the anti-HA affinity matrix (*Figure 2A*). Importantly, however, no shifted band corresponding to SUMO1-conjugated synapsin1 was observed in the His$_6$-HA-SUMO1 KI eluate. Next, we conducted the reverse IP experiment using anti-synapsin1 antibodies to enrich synapsin1 from WT and His$_6$-HA-SUMO1 KI brains, along with anti-IgG IP as a negative control (*Figure 2B*). SDS-PAGE (10%) and Western blot analysis using an anti-synapsin1 antibody confirmed that synapsin1 can be robustly enriched from both WT and His$_6$-HA-SUMO1 KI material with this approach, but again no shifted band indicating SUMO1-synapsin1 was visible using either anti-synapsin1 or anti-HA antibodies (*Figure 2B*). In a final experiment, we pursued a cell-based overexpression approach (*Figure 2C*). GFP-synapsin1a along with HA-SUMO1 were overexpressed in HEK293 cells, and cell lysates were subjected to anti-HA IP (*Figure 2C*). SDS-PAGE (10%) and Western blot analysis of inputs and eluate fractions revealed a general enrichment of HA-SUMO1-conjugated protein species, but synapsin1a was not enriched in the HA eluate fractions (*Figure 2C*). On aggregate, these data indicate that synapsin1a is not a SUMO1-conjugation substrate in vivo, although the possibility remains that SUMO1-conjugated synapsin1a is too transient, unstable, or rare to be detected reliably with our methodology.

## No evidence for SUMO1-conjugation of Gephyrin in vivo and in vitro

Gephyrin, a postsynaptic scaffold protein at inhibitory synapses, was recently reported to be a novel SUMO1 substrate, with additional data indicating that overexpression of components of the SUMOylation machinery modulates synaptic gephyrin clustering (*Ghosh et al., 2016*). We first used the His$_6$-HA-SUMO1 KI model to test whether SUMO1-conjugation of gephyrin is detectable in vivo (*Figure 3*). SDS-PAGE (4–12%) and Western blot analysis of input and eluate fractions from anti-HA IP from WT and His$_6$-HA-SUMO1 KI brain lysates showed that gephyrin binds non-specifically to the affinity matrix (*Figure 3A*). However, we did not detect a gephyrin band shifted by ~20 kDa that would be expected for SUMO1-conjugated gephyrin (*Figure 3A*). Second, we

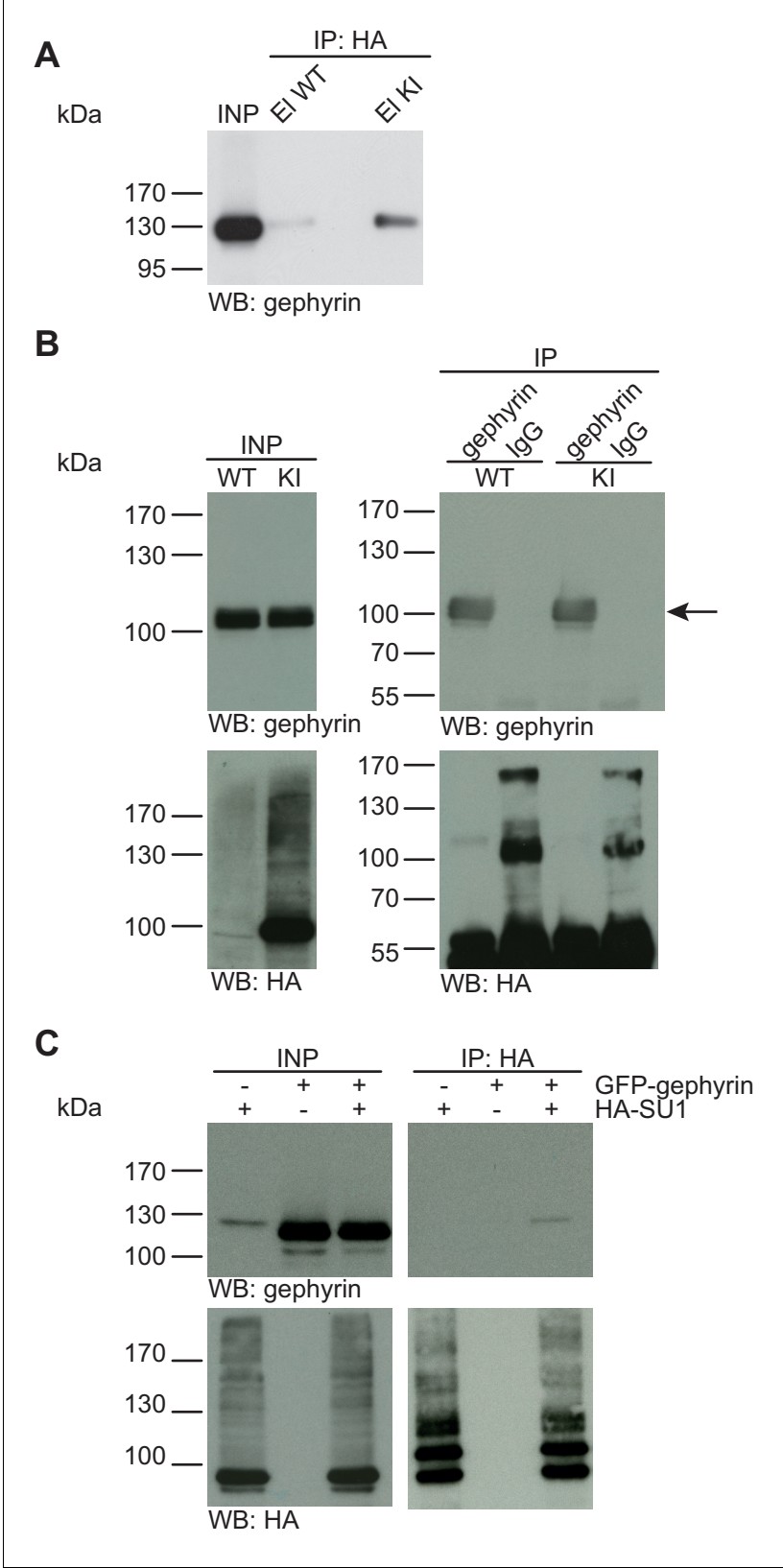

**Figure 3.** Gephyrin is not SUMO1-conjugated in vivo and in vitro. (**A**) SDS-PAGE (4–12%) followed by anti-gephyrin Western blot analysis of input and HA peptide eluate fractions from anti-HA immunoprecipitation in the presence of 20 mM NEM from WT and His$_6$-HA-SUMO1 KI brains. The presence of gephyrin in both WT and KI eluates

*Figure 3 continued on next page*

*Figure 3 continued*
indicates non-specific binding of gephyrin to the affinity matrix. (B) Representative SDS-PAGE (10%) followed by Western blot analysis of input and eluate fractions of anti-gephyrin and anti-IgG immunopurifications in the presence of 20 mM NEM from WT and His$_6$-HA-SUMO1 KI brains. Anti-gepyhrin Western blot confirms the enrichment of gephyrin in both WT and KI brains after affinity purification using anti-gephyrin antibody but not when using mouse IgG (black arrow, upper panel). Importantly, anti-HA Western blot analysis does not reveal SUMO1-gephyrin bands. (C) SDS-PAGE (10%) followed by Western blot analysis of input and eluate fractions of anti-HA immunoprecipitation in the presence of 20 mM NEM from HEK cells overexpressing HA-SUMO1 and GFP-gephyrin, alone or in combination. Anti-HA Western blot analysis confirms the enrichment of HA-SUMO1 conjugates (lower panel) but no SUMO1-gephyrin signal is observed in the eluate fractions (upper panel). Images are representatives of at least three independent experiments.

performed the reverse IP and enriched gephyrin from WT and His$_6$-HA-SUMO1 KI brain lysates, using anti-gephyrin antibodies and IgG as a negative control (*Figure 3B*). SDS-PAGE (10%) and Western blot analysis of the input and eluate fractions using anti-gephyrin and anti-HA antibodies showed that gephyrin was successfully enriched, but no shifted gephyrin bands indicative of SUMO1-conjugation were observed (*Figure 3B*). To examine whether the expression levels of gephyrin and/or SUMO1 could have been a limiting factor in our IP experiments with mouse tissue, GFP-gephyrin and HA-SUMO1 were coexpressed in HEK293 cells, SUMOylated proteins were then enriched from cell lysates by IP using an anti-HA antibody, and the input and IP samples were analysed by SDS-PAGE (10%) and Western blotting using anti-HA and anti-gephyrin antibodies (*Figure 3C*). Once again, no SUMO1-conjugation of gephyrin was apparent. Overall, these data do not provide evidence for SUMO1-conjugation of gephyrin in vivo or in vitro, although the possibility remains that SUMO1-conjugated gephyrin is too transient, unstable, or rare to be detected reliably with our methodology.

## No evidence for SUMO1-conjugation of GluK2 in vivo and in vitro

The kainate receptor subunit GluK2 was the first synaptic protein to be described as a SUMO1 substrate (*Martin et al., 2007*), and the corresponding data triggered a number of follow-up studies by various groups (*Zhu et al., 2012*; *Wilkinson et al., 2012*; *Sun et al., 2014*). Based on the experimental design described in the chapters above, we studied a possible SUMO1-conjugation of GluK2 in vivo and in vitro (*Figure 4*). Anti-HA IP from WT and His$_6$-HA-SUMO1 KI brain lysates followed by SDS-PAGE (4–12%) Western blot analysis of input and IP eluate fractions showed that some GluK2 binds non-specifically to the affinity matrix. However, no shifted GluK2 band indicative of SUMO1-conjugation was detected (*Figure 4A*). Next, we performed anti-GluK2 IP from WT and His$_6$-HA-SUMO1 KI brain, using anti-IgG IP as a negative control, and analysed the input and eluate fractions by SDS-PAGE (10%) and Western blotting with anti-HA and anti-GluK2 antibodies (*Figure 4B*). While GluK2 was strongly enriched in the specific IP fraction, no indication of a shifted band indicative of SUMO1-conjugated GluK2 was obtained with either anti-HA or anti-GluK2 antibodies (*Figure 4B*). We next performed IP experiments upon protein overexpression in HEK293 cells as described in the chapters above (*Figure 4C*). SDS-PAGE (10%) and Western blot analysis of input and eluate fractions after IP did not indicate an enrichment of SUMO1-conjugated GluK2 (*Figure 4C*). On aggregate, these data indicate that GluK2 is either not conjugated to SUMO1, or that its SUMOylation status is too transient, unstable, or rare to be detected reliably in vivo or in vitro.

## No evidence for SUMO1-conjugation of RIM1, Syntaxin1a, Synaptotagmin1, or mGluR7 in vivo

RIM1 is a presynaptic active zone protein that is involved in the control of presynaptic Ca$^{2+}$-channel density, synaptic vesicle priming, and synaptic plasticity (*Calakos et al., 2004*; *Castillo et al., 2002*; *Chevaleyre et al., 2007*; *Han et al., 2015*; *Kaeser et al., 2008*; *Schoch et al., 2002*). RIM1 was recently reported as a SUMO1 substrate, with additional data indicating that SUMO1-conjugation of RIM1 is required for RIM1-mediated Ca$^{2+}$ channel clustering (*Girach et al., 2013*). We investigated the SUMO1-conjugation of RIM1 in vivo using the experimental design described in

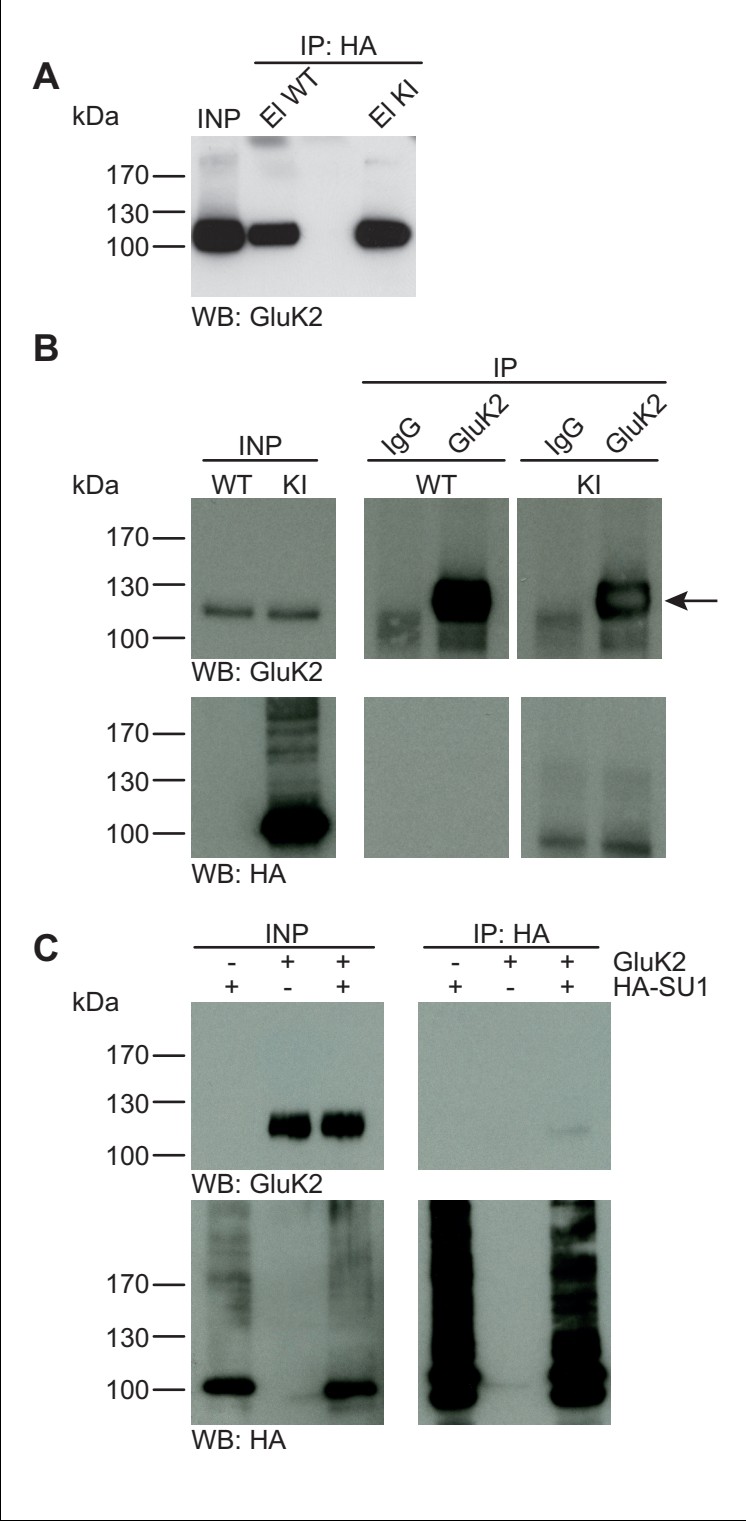

**Figure 4.** GluK2 is not SUMO1-conjugated in vivo and in vitro. (**A**) SDS-PAGE (4–12%) followed by Anti-GluK2 Western blot analysis of input and HA peptide eluate fractions from anti-HA immunoprecipitation in the presence of 20 mM NEM from WT and His$_6$-HA-SUMO1 KI brain. The presence of GluK2 in both WT and KI eluates indicates non-specific binding of GluK2 to the affinity matrix. (**B**) Representative SDS-PAGE (10%) followed by anti-HA and anti-GluK2 Western blot analysis of input and eluate fractions from anti-GluK2 and anti-IgG immunoprecipitations in the presence of 20 mM NEM from WT and His$_6$-HA-SUMO1 KI brains. GluK2 is specifically enriched in both WT and KI samples after affinity purification using anti-GluK2 but not in control

*Figure 4 continued on next page*

*Figure 4 continued*
experiment conducted with rabbit IgG (upper panel). However, anti-HA Western blot does not reveal any band that could have corresponded to SUMO1-GluK2 (lower panel). (C) Representative SDS-PAGE (10%) followed by anti-HA and anti-GluK2 Western blot of input and eluate fractions from anti-HA immunoprecipitation in the presence of 20 mM NEM from HEK cells overexpressing HA-SUMO1 and GluK2, alone or in combination. Anti-HA Western blot analysis confirms the enrichment of HA-SUMO1 conjugates (lower panel) but no SUMO1-GluK2 band is observed in the eluate fractions (upper panel). Images are representatives of at least three independent experiments.

the chapters above (*Figure 5A*). First, anti-HA IP from WT and His$_6$-HA-SUMO1 KI brain lysates, followed by SDS-PAGE (4–12%) and Western blot analysis revealed that RIM1 binds non-specifically to the affinity matrix. However, no SUMO1-conjugated RIM1 variant with a correspondingly shifted molecular weight was detected in the His$_6$-HA-SUMO1 KI sample (*Figure 5A*). Next, we performed anti-RIM1 IP from WT and His$_6$-HA-SUMO1 KI brain, along with anti-IgG IP as a negative control, and analysed the input and eluate fractions by SDS-PAGE (8%) and Western blot using anti-HA and anti-RIM1 antibodies (*Figure 5B*). While RIM1 was strongly enriched in the specific IP sample, no SUMO1-conjugated RIM1 band with correspondingly shifted molecular weight was detectable with anti-HA or anti-RIM1 antibodies (*Figure 5B*). With the same methodological approach, we next determined the SUMO1-conjugation status of the SNARE protein syntaxin1a in vivo (*Figure 5C and D*) because SUMO1-conjugation of syntaxin1a had recently been reported (*Craig et al., 2015*). As seen for some of the other proteins tested, SDS-PAGE (4–12%) and Western blot analysis of input and eluate fractions from WT and His$_6$-HA-SUMO1 KI samples revealed non-specific syntaxin1a binding to the affinity matrix, but again, no syntaxin1a band with a molecular weight increase of ~20 kDa was detected in the His$_6$-HA-SUMO1 KI sample (*Figure 5C*). In the reverse IP experiment, syntaxin1a was clearly and specifically enriched from WT and KI brains with anti-syntaxin1a antibodies (*Figure 5D*), but no anti-HA immunoreactive band representing SUMO1-conjugated syntaxin1a was detectable (higher molecular weight bands seen in the blot with anti-syntaxin1a antibodies likely represent SNARE complexes and/or IgG from the IP). In a final set of experiments, we employed our IP strategy with His$_6$-HA-SUMO1 KI and WT control material to assess the SUMO1-conjugation of synatoptagmin1 and mGluR7 (*Matsuzaki et al., 2015*; *Choi et al., 2016*). Here again, we did not detect SUMO1-conjugated protein variants with a correspondingly shifted molecular weight (*Figure 5E*). In sum, our biochemical analyses of His$_6$-HA-SUMO1 KI material did not provide evidence for the SUMO1-conjugation of RIM1, syntaxin1a, synaptotagmin1, or mGluR7 in vivo.

## Specific SUMO1 substrates are absent from biochemically enriched synaptic fractions of mouse brain

In a more general attempt to assess the abundance of synaptic SUMO1-conjugates, we used biochemical methods to identify SUMO1-conjugated proteins in subcellular synaptic fractions from mouse brain without focusing on any particular substrate protein. Previous reports indicated the presence of multiple SUMO1-conjugates in synaptosome fractions prepared from mouse or rat brain (*Martin et al., 2007*; *Luo et al., 2013*), leading to the proposal that SUMO1-conjugated proteins are present at synapses. To test this notion, we sub-fractionated cortices of His$_6$-HA-SUMO1 KI and SUMO1 KO mice by standard sub-fractionation protocols. SDS-PAGE (4–12%) and Western blot analysis of the different fractions using anti-GluN1 and anti-synaptophysin antibodies confirmed that the fractionation procedure worked properly (*Figure 6*, bottom panels). We then assessed the levels of SUMO1-conjugates in the various fractions by SDS-PAGE and anti-SUMO1 or anti-HA Western blotting (*Figure 6*, upper panels). With this approach, the comparison of immunoblots from His$_6$-HA-SUMO1 KI and SUMO1 KO brains allows to determine which bands correspond specifically to SUMO1-conjugated proteins - bands that appear in both the His$_6$-HA-SUMO1 KI and SUMO1 KO samples are false-positives (*Figure 6*, asterisks). As expected, the anti-SUMO1 signal was substantially reduced in the total homogenate (H) sample from SUMO1 KO brain as compared to His$_6$-HA-SUMO1 KI brain (*Figure 6*, top panel). SUMO1 conjugated proteins appear typically as a smear of protein bands above 90 kDa (*Figure 6*, brackets) and as a prominent band at ~90 kDa, which

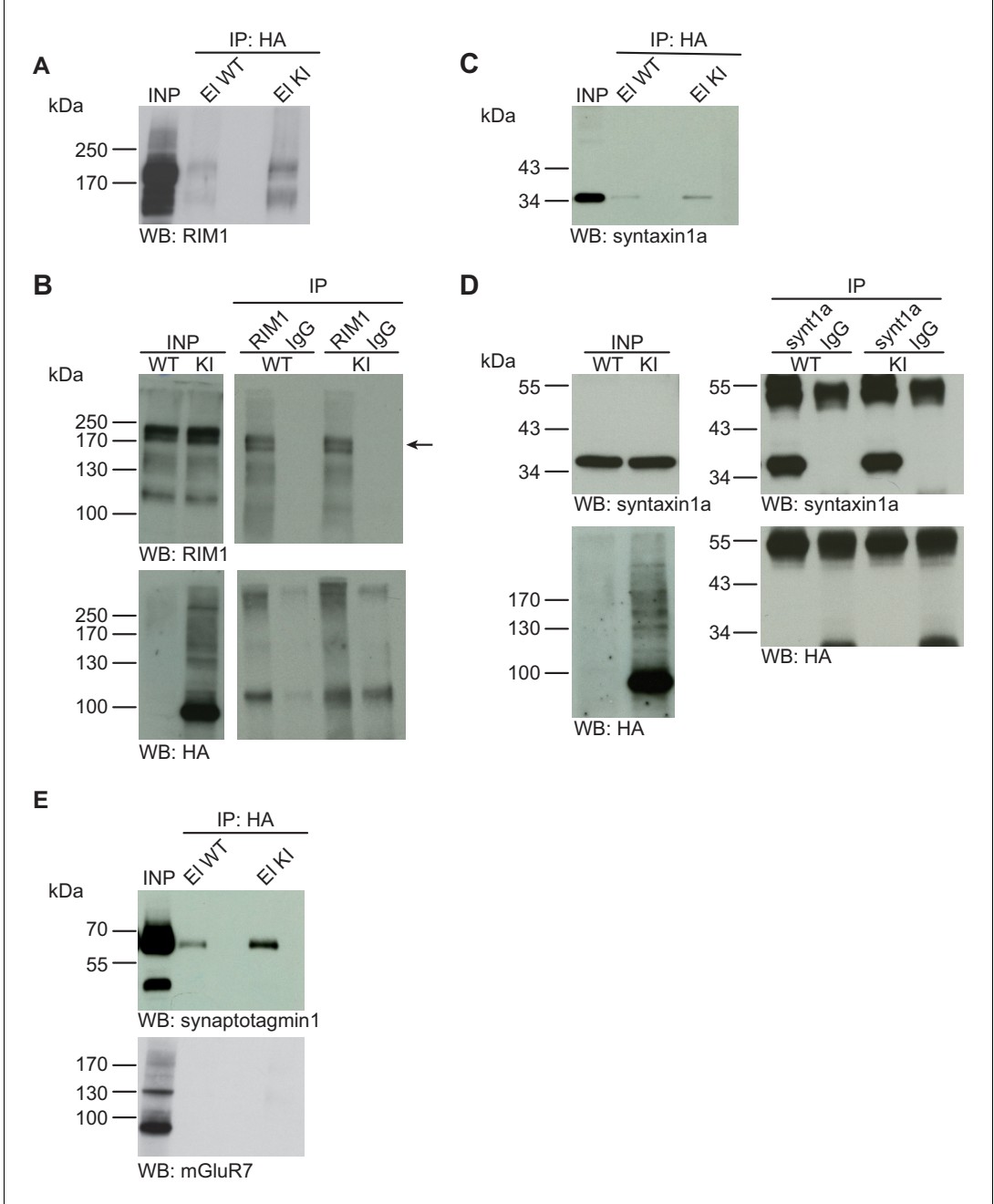

**Figure 5.** Syntaxin1, RIM1, SynaptotagminI, and mGluR7 are not SUMO1-conjugated in vivo. (A) SDS-PAGE (4–12%) followed by Western blot analysis using anti-RIM1 antibody of input and HA peptide eluate fractions from anti-HA immunoprecipitation in the presence of 20 mM NEM from WT and His$_6$-HA-SUMO1 KI brains. The presence of RIM1 in both WT and KI eluates indicates non-specific binding of RIM1 to the affinity matrix (B) Representative SDS-PAGE (8%) followed by Western blot analysis of input and eluate fractions of anti-RIM1 and anti-IgG immunopurifications in the presence of 20 mM NEM from WT and His$_6$-HA-SUMO1 KI brains. Western blot analysis using anti-RIM1 confirms the enrichment of RIM1 in both WT and KI samples solely when anti-RIM1 antibody is used (upper panel). However, no SUMO1-RIM1 band is apparent (lower panel). (C) SDS-PAGE (4–12%) followed by anti-syntaxin1α Western blot analysis of input and HA peptide eluate fractions from anti-HA immunoprecipitation in the presence of 20 mM NEM from WT and His$_6$-HA-SUMO1 KI brains. The presence of syntaxin1a in both WT and KI eluates indicates non-specific binding of syntaxin1a to the affinity matrix (D) Representative SDS-PAGE (12%) followed by Western blot analysis of input and eluate fractions of anti-syntaxin1α and anti-IgG immunopurifications in the presence or absence of 20 mM NEM from WT and His$_6$-HA-SUMO1 KI brains. Western blot analysis using anti-syntaxin1a confirms the enrichment of syntaxin1a in both WT and KI samples solely when anti-syntaxin1a antibody is used (upper panel). However, no SUMO1-syntaxin1a band is apparent (lower panel). (E) SDS-PAGE (4–12%) followed by anti-synaptotagmin1 and mGluR7 Western blot of input and anti-HA peptide eluate fractions from anti-HA immunoprecipitation in the presence of 20 mM NEM from WT and His$_6$-HA-SUMO1 KI brains. The presence of

*Figure 5 continued on next page*

Figure 5 continued
synaptotagmin1 in both WT and KI eluates indicates non-specific binding of synaptotagmin1 to the affinity matrix, while mGluR7 is not enriched in either case. Images are representatives of at least three independent experiments.

represents SUMO1-conjugated RanGAP1 (*Figure 6*), the most abundant SUMO1-conjugated protein known (*Tirard et al., 2012*; *Mahajan et al., 1997*; *Zhang et al., 2008*). This typical band pattern is observed in His₆-HA-SUMO1 KI samples with either anti-HA or anti-SUMO1 antibodies, particularly in the unfractionated homogenate (H), and in the nuclear (P1) and cytosolic fractions (S2). Synaptic fractions (i.e. LP1, LP2, and SPM), however, contain very few specific anti-HA-positive protein bands, with the exception of some RanGAP1 contamination at ~90 kDa (*Figure 6*). Anti-SUMO1 blotting of these brain fractions yielded a band pattern that is overall similar to the one seen with anti-HA label-ing. In addition, anti-SUMO1 blotting labelled multiple additional bands, also in the synaptic

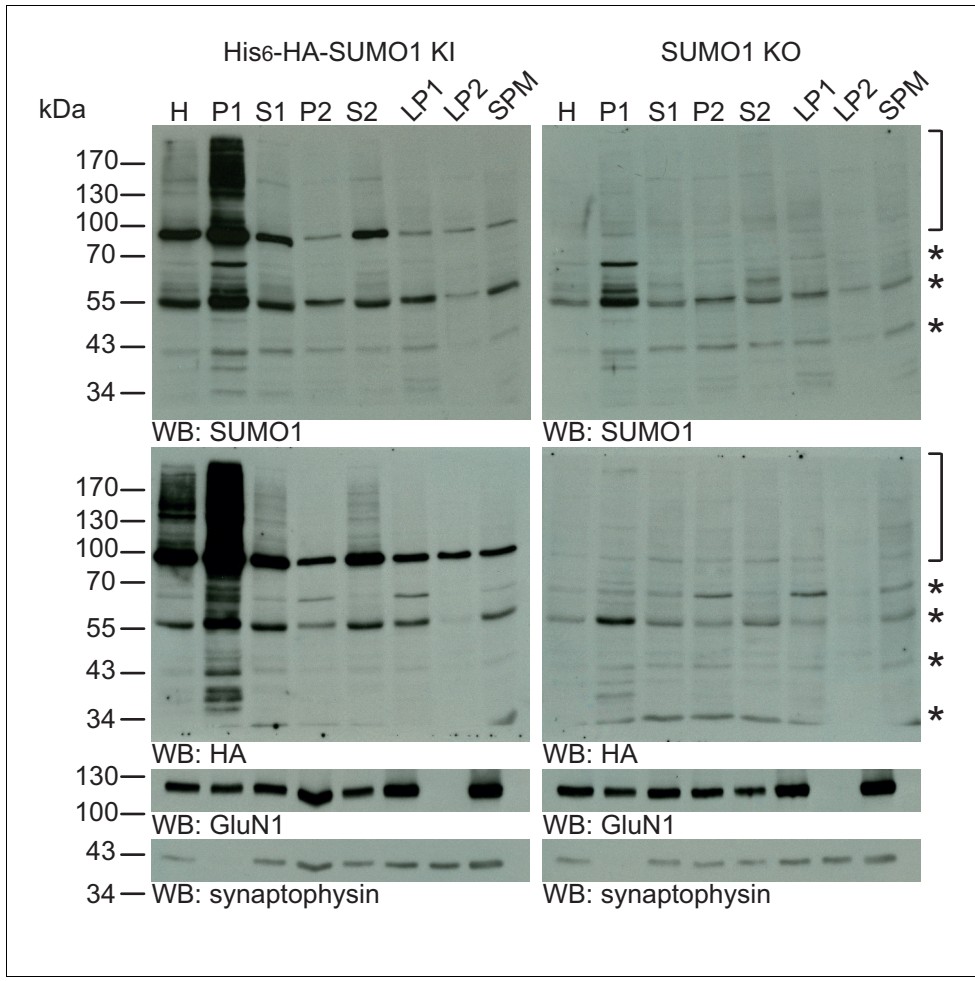

**Figure 6.** SUMO1 conjugates are mainly in nuclear fractions. SDS-PAGE (4–12%) followed by Western blot analysis of subcellular fractions performed in the presence of 20 mM NEM of His₆-HA-SUMO1 KI and WT mouse brain using anti-SUMO1 (upper two panels) and anti-HA antibodies (below), and antibodies to GluN1 and synaptophysin to validate the fractionation procedure (lower two panels). Bracket indicates specific SUMO1 substrates while stars indicate non-specific SUMO1 bands present in both His₆-HA-SUMO1 KI and SUMO1 KO. H, homogenate; P1, nuclear pellet; S1, supernatant after P1 sedimentation; P2, crude synaptosomal pellet; S2, supernatant after P2 sedimentation; LP1, lysed synaptosomal membranes; LS1, supernatant after LP1 sedimentation; SPM, synaptic plasma membranes. Images are representatives of at least three independent experiments.

fractions, which were also detected in the SUMO1 KO samples, demonstrating that these bands are false-positives (*Figure 6*, asterisks). On aggregate, these data indicate that specific SUMO1-conjugated proteins cannot be detected in synaptic fractions by using a very sensitive Western blotting approach.

## Immunolabelling shows no evidence for SUMO1 localisation at pre- or postsynaptic sites in cultured hippocampal neurons

### Anti-HA immunolabelling shows no SUMO1 at synapses in His$_6$-HA-SUMO1 KI neurons

We next employed an immunofluorescence labelling approach to assess the presence of SUMO1 and SUMO1-conjugates at neuronal synapses. This approach has been used previously by us and others, but the results have been contradictory (*Tirard et al., 2012*; *Girach et al., 2013*; *Martin et al., 2007*; *Konopacki et al., 2011*). In our attempts to explain this discrepancy, we noticed that the authors of several relevant studies chose to use digitonin for cell permeabilisation prior to immunostaining, instead of the more commonly used Triton X-100. This choice was based on the notion that digitonin preferentially permeabilises the plasma membrane, hence reduces the contribution of the nucleus to overall staining, and possibly unmasks synaptic SUMO1 staining (*Girach et al., 2013*; *Craig et al., 2015*, *2012*; *Jaafari et al., 2013*).

To examine whether digitonin permeabilisation reveals synaptic SUMO1, we performed anti-HA immunostaining of mature hippocampal neurons from His$_6$-HA-SUMO1 KI and SUMO1 KO neurons in parallel using three different permeabilisation protocols - a standard Triton X-100 method and permeabilisation with two different digitonin concentrations (20 and 10 µg/ml), which had both been used previously to immunostain cultured neurons with antibodies against SUMO1 (*Girach et al., 2013*; *Craig et al., 2015*; *Konopacki et al., 2011*; *Jaafari et al., 2013*). Strikingly, using all three permeabilisation methods, we observed strong immunolabelling of His$_6$-HA-SUMO1 inside the nucleus and at the nuclear envelope (*Figures 7*, *8* and *9*) while SUMO1 KO neurons exhibited no anti-HA immunolabelling in the nucleus or nuclear envelope. These data demonstrate that both digitonin (at 20 and 10 µg/ml) and Triton X-100 permeabilise the nucleus sufficiently to allow for SUMO1 labelling of nuclear structures. For subsequent anti-SUMO1 immunolabelling studies, we followed the fixation, permeabilisation, and blocking protocol described by *Jaafari et al. (2013)*, which represents the most comprehensive published protocol for anti-SUMO1 immunolabelling after digitonin permeabilisation.

No specific colocalisation between anti-synapsin and anti-HA immunolabelling was observed under any of the permeabilisation conditions that we tested (*Figure 7*, *8* and *9*, white arrowhead). Indeed, little specific anti-HA immunolabelling was observed outside of the nucleus of His$_6$-HA-SUMO1 KI neurons. Additionally, Map2 staining was not homogenous along neuronal somata and dendrites after permeabilisation with 10 µg/ml digitonin, indicating that this digitonin concentration leads to incomplete permeabilisation and/or provides only restricted access to some antigens (*Figure 9*).

Overall, based on anti-HA immunolabelling performed with His$_6$-HA-SUMO1 KI and SUMO1 KO neurons, we found no evidence for the localisation of SUMO1 or SUMO1-conjugates to synaptic structures, regardless of the permeabilisation method used prior to immunolabelling.

### Anti-SUMO1 immunolabelling shows no SUMO1 at synapses in His$_6$-HA-SUMO1 KI neurons

It was argued previously that the His$_6$-HA tag added to SUMO1 in the His$_6$-HA-SUMO1 KI might prevent SUMO1 from modifying key substrates or that the reduction of global SUMO1 levels in His$_6$-HA-SUMO1 KI that we previously reported and described above (*Figure 1—figure supplement 1*) may account for the absence of SUMO1 signal at synapses in His$_6$-HA-SUMO1 KI neurons as compared to WT samples (*Tirard et al., 2012*; *Henley et al., 2014*; *Luo et al., 2013*). Therefore, we decided to evaluate how anti-SUMO1 immunolabelling differed from the anti-HA labelling observed in the His$_6$-HA-SUMO1 KI (*Figures 7*, *8* and *9*), especially in the synaptic compartment. To this end, we used an antibody against SUMO1 itself, which had been used in various studies to detect SUMO1 at synapses (monoclonal anti-SUMO1 clone 21C7; see Western blots in *Figure 6*), with SUMO1 KO neurons as negative controls to assess the specificity of the anti-SUMO1 immunostaining

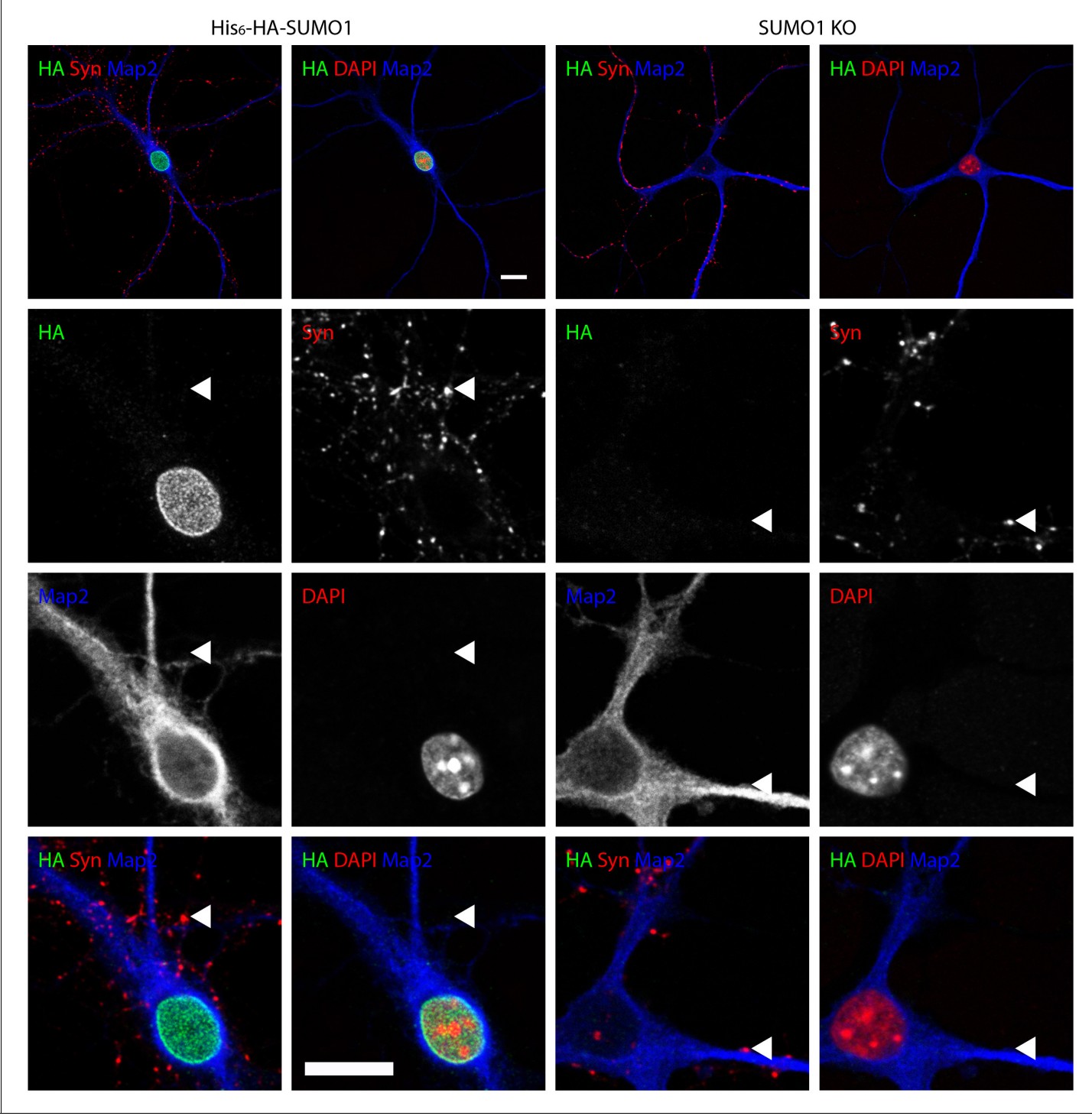

**Figure 7.** Anti-HA immunolabelling in His$_6$-HA-SUMO1 KI neurons permeabilised with Triton X-100 is mainly nuclear. Hippocampal neurons from either His$_6$-HA-SUMO1 KI mice or SUMO1 KO mice were fixed in 4% PFA and permeabilised using 0.3% Triton X-100, 10% goat serum, and 0.1% fish skin gelatin for 20 min. Primary antibodies were 1:1000 mouse anti-HA (green), 1:2000 rabbit anti-synapsin 1/2 (red), and 1:500 chicken anti-Map2 (blue). In His$_6$-HA-SUMO1 KI cells, anti-HA labelling was primarily observed in nuclei. Faint anti-HA labelling and occasional puncta were also observed, which did not generally correspond with synapsin-positive puncta (arrowheads). Data representative of 2 independent experiments. Scale bars, 10 μm.

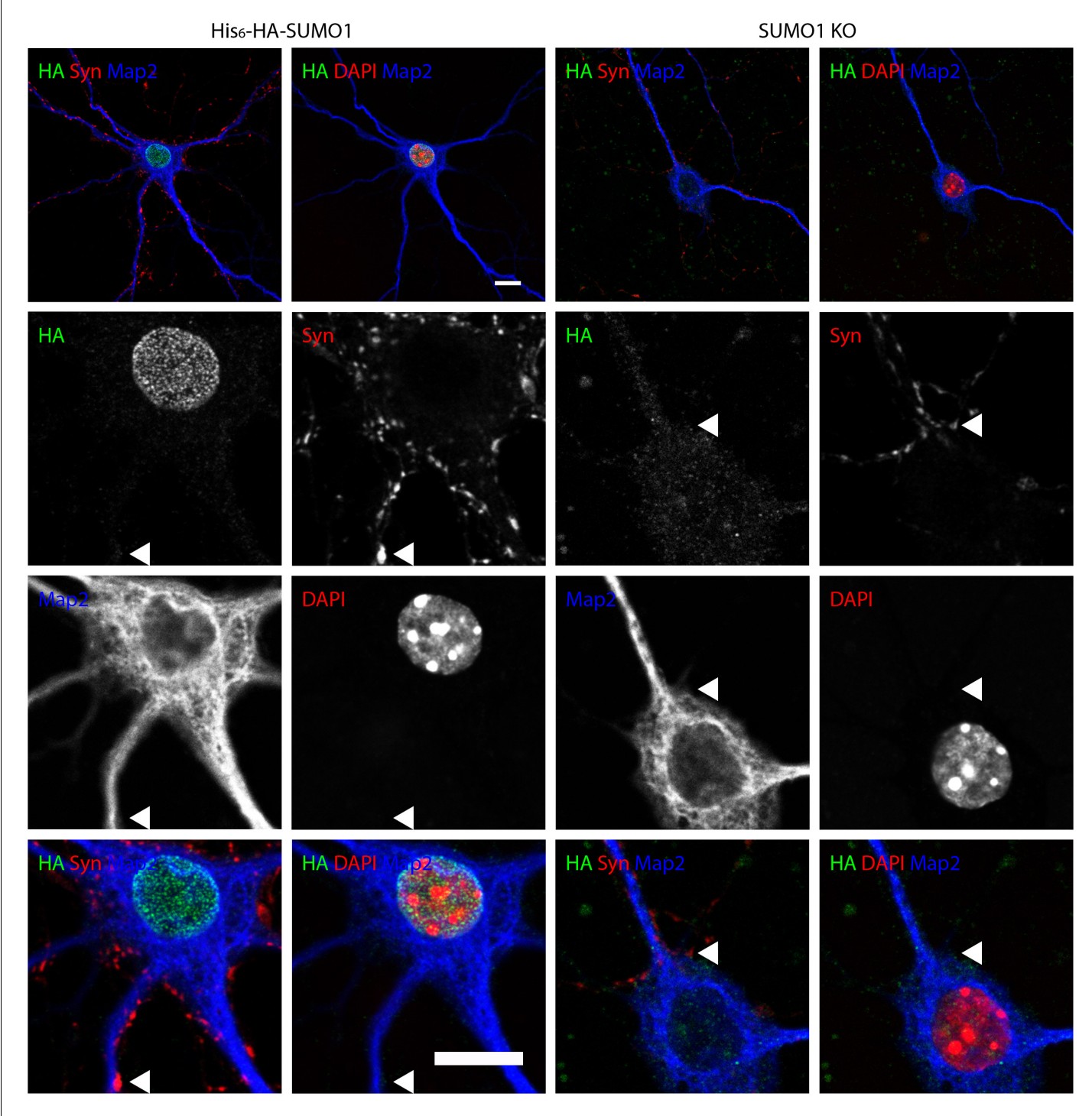

**Figure 8.** Anti-HA immunolabelling in His$_6$-HA-SUMO1 KI neurons permeabilised with 20 µg/ml digitonin is mainly nuclear. Hippocampal neurons from either His$_6$-HA-SUMO1 KI mice or SUMO1 KO mice were fixed in 4% PFA and permeabilised using 20 µg/ml digitonin for 20 min, then blocked with 10% horse serum for 20 min. Primary antibodies were 1:1000 mouse anti-HA (green), 1:2000 rabbit anti-synapsin 1/2 (red), and 1:500 chicken anti-Map2 (blue). In His$_6$-HA-SUMO1 KI cells, anti-HA labelling was primarily observed in nuclei. Faint anti-HA labelling and occasional puncta were also observed, which did not generally correspond with synapsin-positive puncta (arrowheads). Data representative of 2 independent experiments. Scale bars, 10 µm.

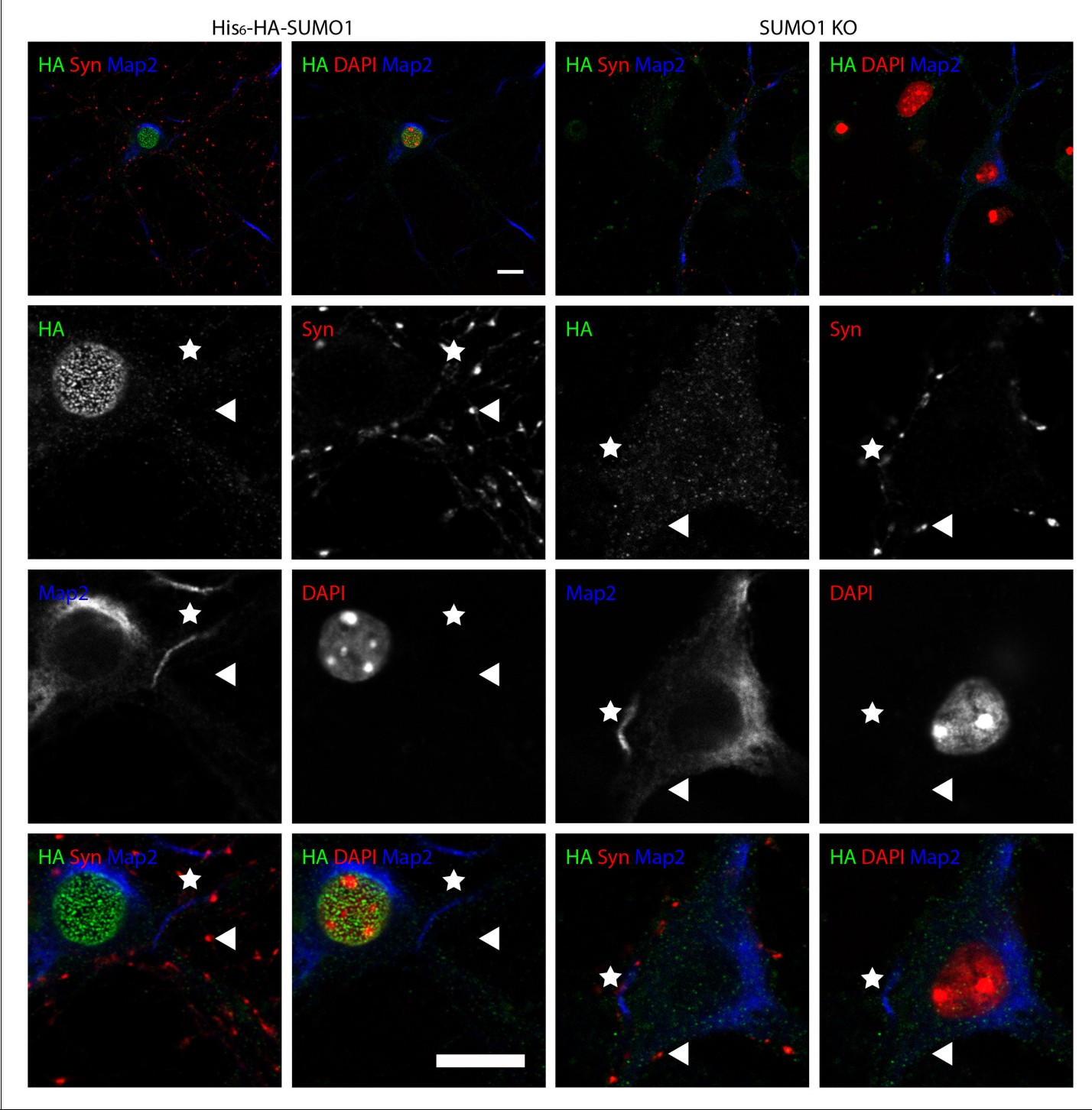

**Figure 9.** Anti-HA immunolabelling in His$_6$-HA-SUMO1 KI neurons permeabilised with 10 µg/ml digitonin is mainly nuclear. Hippocampal neurons from either His$_6$-HA-SUMO1 KI mice or SUMO1 KO mice were fixed in 4% PFA and permeabilised using 10 µg/ml digitonin, then blocked with 10% horse serum for 10 min. Primary antibodies were 1:1000 mouse anti-HA (green), 1:2000 rabbit anti-synapsin 1/2 (red), and 1:500 chicken anti-Map2 (blue). In His$_6$-HA-SUMO1 KI cells, anti-HA labelling was primarily observed in nuclei. Faint anti-HA labelling and occasional puncta were also observed, which did not generally correspond with synapsin-positive puncta (arrowheads). Permeabilisation with 10 µg/ml digitonin resulted in incomplete immunolabelling with Map2, with dendrites appearing 'fragmented' (indicated by stars) as compared to samples permeabilised with Triton X-100 or 20 µg/ml digitonin. Data representative of 2 independent experiments. Scale bars, 10 µm.

(*Figures 10* and *11*). With Triton X-100 permeabilisation (*Figure 10*), anti-SUMO1 labelling was observed in the nucleus of all neurons, with minimal staining in the surrounding cytoplasm (*Figure 10*, white arrow). Anti-SUMO1 immunolabelling in the nuclei of neurons from SUMO1 KO mice was greatly reduced, as expected (*Figure 10*, white arrow). When neurons from the same animals were permeabilised with 10 µg/ml digitonin instead of Triton X-100, nuclear anti-SUMO1 immunolabelling was virtually absent (*Figure 11*). This observation supports previous reports, which indicated that digitonin reduces nuclear SUMO1 immunolabelling by antibodies against SUMO1 (see Figure 16 for further analysis) (*Girach et al., 2013*; *Craig et al., 2015*; *Konopacki et al., 2011*; *Jaafari et al., 2013*; *Kantamneni et al., 2011*). However, using either Triton X-100- or digitonin-based methods of permeabilisation, there appeared to be no specific anti-SUMO1 immunolabelling at synapsin-positive puncta (*Figures 10* and *11*, arrowheads). While some punctate immunolabelling was observed in the dendrites of neurons, this was evident in both His$_6$-HA-SUMO1 KI and SUMO1 KO neurons. In sum, neither anti-HA nor anti-SUMO1 antibodies detect specific SUMO1 signals at synapses. However, the lack of synaptic anti-SUMO1 immunolabelling might have been attributable to the reduction in SUMO1-conjugation levels observed in His$_6$-HA-SUMO1 KI mouse brain (*Tirard et al., 2012*) and cultured neurons (*Figure 1—figure supplement 1*). Hence, the data described above do not exclude the possibility that the absence of synaptic His$_6$-HA-SUMO1 signal is an indirect consequence of the effect of the His$_6$-HA-tag on global SUMO1-conjugation levels. Consequently, we further characterized anti-SUMO1 signals at synapses in WT neurons as compared to SUMO1 KO neurons.

## Quantification of anti-SUMO1 immunolabelling does not indicate SUMO1 at synapses in WT neurons

At this point of our analysis, the possibility remained that some specific anti-SUMO1 immunolabelling is localised to synapses but could not be detected using the largely qualitative methods employed thus far. Accordingly, we took a quantitative approach to the analysis of anti-SUMO1 labelling at synapses, under conditions of Triton X-100 and digitonin permeabilisation. In light of the moderate reduction in anti-SUMO1 immunolabelling evident in His$_6$-HA-SUMO1 KI neurons as compared to WT cells (*Figure 1—figure supplement 1*), we compared WT with SUMO1 KO neurons. For this analysis, hippocampal neurons were cultured from pairs of neonatal littermates, in which each pair consisted of a WT pup and an SUMO1 KO pup. Mature neurons from all pairs were then fixed, permeabilised, and immunolabelled concurrently, minimizing any inter-sample differences in experimental conditions. Immunolabelling with an antibody against synapsin1 was used to define presynaptic sites (*Figures 12* and *13*) and postsynaptic structures were defined by labelling with an antibody against shank2 (*Figures 14* and *15*). Neurons were then imaged and the degree of colocalisation between either anti-synapsin or anti-shank2 and anti-SUMO1 immunolabelling was quantified using methods and statistical rationale as described in Materials and methods.

First, in digitonin-permeabilised immunolabelled hippocampal neurons from WT mice, almost half of all synapsin-positive pixels colocalised with SUMO1-positive pixels (Manders M1; *Figure 12C*). Strikingly, however, this was also true of neurons from SUMO1 KO mice, indicating that while a significant proportion of presynaptic sites are immunolabelled with the anti-SUMO1 antibody, the antibody against SUMO1 binds non-specifically to synaptic structures to such an extent that there is no detectable difference between neurons that express SUMO1 and neurons that lack it completely (*Figure 12C*). This was also seen when the proportion of SUMO1-positive pixels that also contain anti-synapsin labelling was quantified (Manders M2; *Figure 12C*). Subsequently, an independent analytical method was applied to the same images (*Figure 12D*). This method allowed quantification of the average intensity of anti-SUMO1 immunolabelling at synapsin-positive puncta, rather than examining the degree of colocalisation between the two antibodies. Here, we found that the average anti-SUMO1 fluorescence intensity at synapsin-positive puncta did not differ between WT and SUMO1 KO neurons (*Figure 12D*), further demonstrating that there is no specific anti-SUMO1 immunolabelling at synapsin-positive sites in WT neurons. The average synapsin labelling intensity and the size of synapsin-positive puncta were also unchanged between WT and SUMO1 KO neurons (*Figure 12D*). The same analysis was carried out in WT and SUMO1 KO neurons that had been permeabilised using Triton X-100 rather than digitonin (*Figure 13*). Again, quantification showed that there is no difference between WT and SUMO1 KO neurons in terms of the intensity of anti-SUMO1

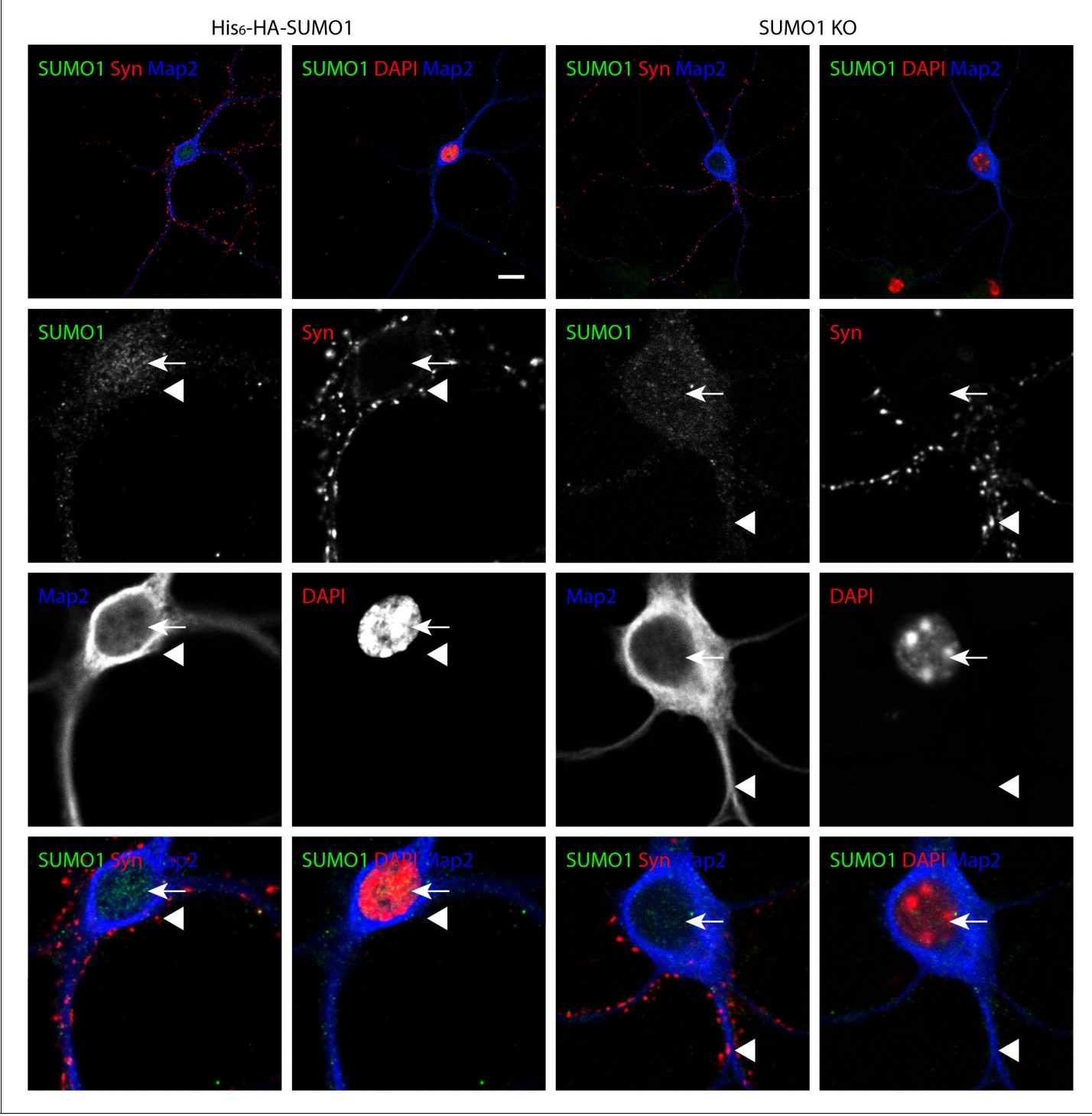

**Figure 10.** Specific anti-SUMO1 immunolabelling in His6-HA-SUMO1 KI neurons permeabilised with Triton X-100 is mainly nuclear and not in synaptic boutons. Hippocampal neurons from either His6-HA-SUMO1 KI mice or SUMO1 KO mice were fixed in 4% PFA and permeabilised using 0.3% Triton X-100. Primary antibodies were 1:50 mouse anti-SUMO1 (green), 1:2000 rabbit anti-synapsin 1/2 (red), and 1:500 chicken anti-Map2 (blue). In His6-HA-SUMO1 KI neurons but not in SUMO1 KO neurons, anti-SUMO1 labelling was observed in nuclei (arrows). Additional anti-SUMO1 labelling was observed similarly in both His6-HA-SUMO1 KI and SUMO1 KO samples, which did not appear to correspond with synapsin-positive puncta (arrowheads). Data are representative of 5 independent experiments. Scale bars, 10 µm.

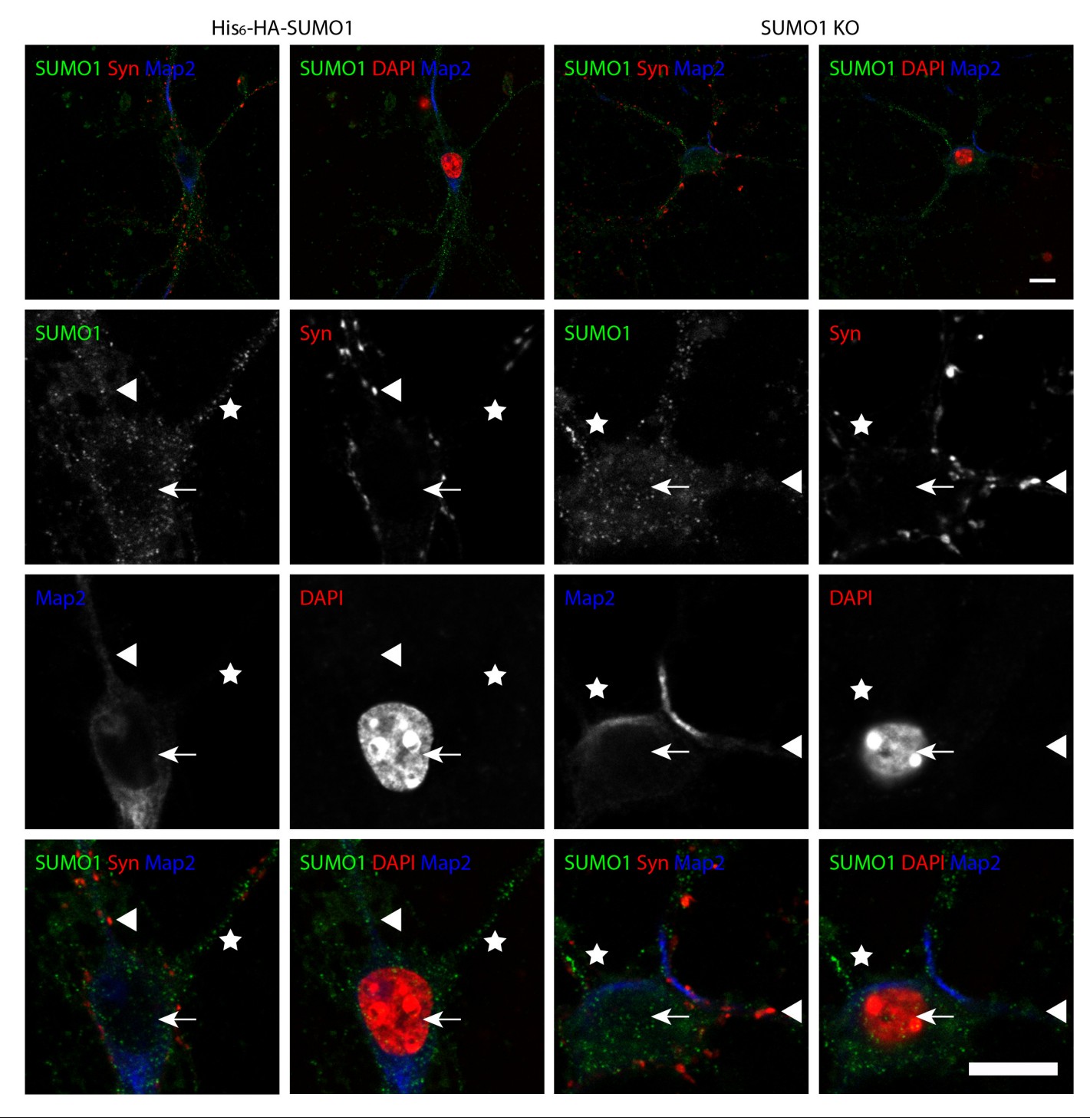

**Figure 11.** Specific anti-SUMO1 immunolabelling in His$_6$-HA-SUMO1 KI neurons permeabilised with 10 µg/ml digitonin does not correspond with synaptic boutons. Hippocampal neurons from either His$_6$-HA-SUMO1 KI mice or SUMO1 KO mice were fixed in 4% PFA and permeabilised using 10 µg/ml digitonin. Primary antibodies were 1:50 mouse anti-SUMO1 (green), 1:2000 rabbit anti-synapsin 1/2 (red), and 1:500 chicken anti-Map2 (blue). In His$_6$-HA-SUMO1 KI neurons anti-SUMO1 labelling was generally absent in nuclei (arrows) as well as in SUMO1 KO neurons. Additional anti-SUMO1 labelling was observed similarly in both His$_6$-HA-SUMO1 KI and SUMO1 KO samples, which did not appear to correspond with synapsin-positive puncta (arrowheads). Permeabilisation with 10 µg/ml digitonin resulted in incomplete immunolabelling with Map2, with dendrites appearing 'fragmented' (indicated by stars). Data are representative of 5 independent experiments. Scale bars, 10 µm.

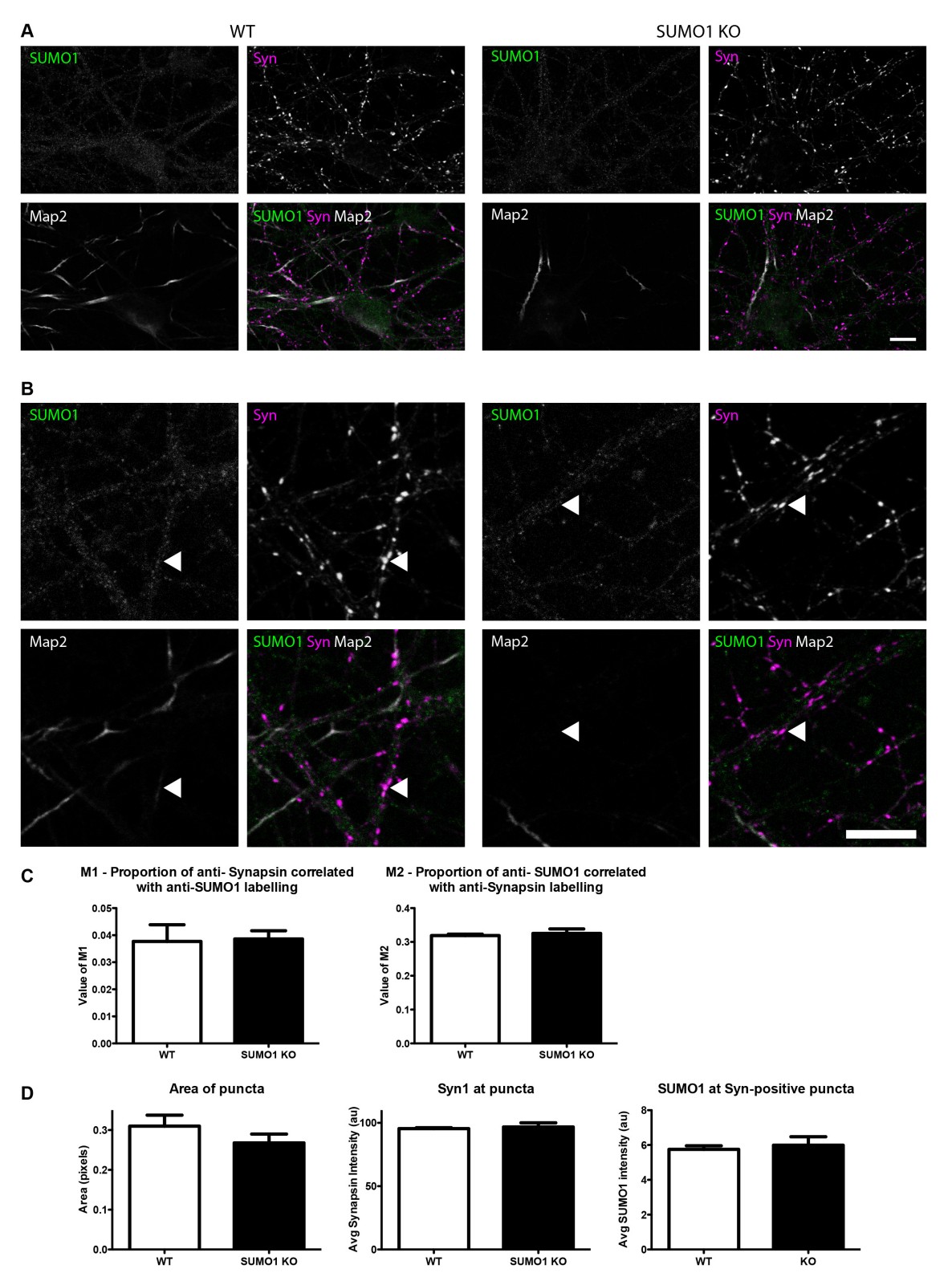

**Figure 12.** Quantification of anti-SUMO1 immunolabelling shows that SUMO1 is not specifically detected at presynaptic sites in digitonin-permeabilised neurons. (**A**) Primary hippocampal neurons from WT and SUMO1 KO littermates were fixed, then digitonin-permeabilised (10 µg/ml), and immunolabelled with antibodies against SUMO1 (green), synapsin (purple), and Map2 (white). A region of interest (neuron) was identified, automatically divided into a grid composed of 6–12 fields of view, and z-stacks were taken for each field. Fields of view were 'stitched' back together to form

*Figure 12 continued on next page*

*Figure 12 continued*

overview images. (**B**) Inset regions from (**A**) are shown in greater detail. (**C**) Manders' correlation coefficients between anti-SUMO1 and anti-synapsin signals were determined. The degree of colocalisation between anti-synapsin and anti-SUMO1 was not different between WT and SUMO1 KO neurons. (**D**) In a separate analysis, synapsin-positive puncta were automatically segmented and the area (left panel) and fluorescence intensity (centre panel) was calculated in synapsin-positive regions of interest (puncta). The intensity of anti-SUMO1 immunolabelling was also quantified at synapsin-positive puncta (right panel). No significant difference between WT and SUMO1 KO neurons was observed for anti-SUMO1 intensity at synapsin-positive puncta. These data indicate that anti-SUMO1 staining at synaptic sites is not specific for SUMO1. Images represent immunostaining of cultures from 3 mice of each genotype. Scale bar, 10 μm. Error bars represent SEM.

immunolabelling at synapsin-positive puncta (*Figure 13C*), and the method of permeabilisation had no significant effect on the intensity and size of synapsin positive puncta (*Figure 13D*). We obtained similar results when colocalisation between anti-SUMO1 and anti-shank2 signals was quantified (*Figure 14*). Using either a colocalisation-based approach (*Figure 14C*) or by quantifying anti-SUMO1 labeling intensity at shank2-positive puncta (*Figure 14D*), there was no detectable difference in colocalisation between WT and SUMO1 KO neurons. Triton-permeabilised neurons also showed no difference in anti-SUMO1 intensity at shank2-positive puncta between WT and SUMO1 KO neurons (*Figure 15*). Thus, neither presynaptic nor postsynaptic sites in cultured hippocampal neurons exhibited specific anti-SUMO1 immunolabelling.

While anti-SUMO1 immunolabelling at synapses was not different between WT and SUMO1 KO neurons, it was greatly reduced in the nucleus of SUMO1 KO neurons when Triton X-100 was used for permeabilisation (*Figure 16A and C*). In cells permeabilised with digitonin, there was no significant difference between nuclear anti-SUMO1 labelling in WT and SUMO1 KO neurons (*Figure 16B and C*). This is broadly consistent with previous data indicating that digitonin permeabilisation largely prevents anti-SUMO1 antibodies from labelling the nucleus (*Girach et al., 2013*; *Craig et al., 2015*; *Jaafari et al., 2013*), although WT neurons exhibited some variability in terms of nuclear anti-SUMO1 labelling. Nuclear labelling was occasionally observed (2 out of 14 neurons, *Figure 16B*, left panels), but absent in the remaining cells (*Figure 16B*, center panels). In neurons where nuclear labelling was absent, a SUMO1-positive perinuclear 'ring' was often evident, likely representing the nuclear envelope where SUMO1 is abundant (*Figure 16B*, centre panels). Both nuclear and perinuclear anti-SUMO1 immunolabelling was absent in SUMO1 KO neurons (*Figure 16B*, right panels). The neurons used for this analysis were the same neurons that were analysed for *Figures 13* and *14*. Overall, these findings demonstrate that nuclear anti-SUMO1 immunolabelling is reduced in SUMO1 KO neurons. This supports the conclusion that specific anti-SUMO1 signals are absent from synapses not because of a failure by the anti-SUMO1 antibody to detect SUMO1 but rather because of a lack of SUMO1 at synaptic sites.

In sum, our immunolabelling data do not provide evidence of the presence of SUMO1 or SUMO1-conjugates at synapses.

## Discussion

Analyses of post-translational protein modifications by SUMOylation are challenging because genuine SUMO-conjugated protein species are typically of low abundance, because physiologically relevant SUMOylation is often transient, and because SUMOylation is very sensitive to proteolytic reversal, particularly also during the experimental processing of biological samples. Despite these challenges, the interest in research concerning SUMOylation in neurons has grown continuously over the past decade, particularly regarding SUMO1-conjugation of proteins that play fundamental roles in synaptic transmission. The emergent view resulting from the corresponding studies is that SUMO1 modification of synaptic proteins is rather prevalent and has profound effects on synaptic transmission (*Henley et al., 2014*; *Schorova and Martin, 2016*). Because of our primary focus on the molecular mechanisms of synaptic transmission, these publications have been of great interest to us, and we have independently been investigating the role of SUMOylation in synapses. To our disappointment, we could not find evidence for SUMO1 modification of seven selected, previously published synaptic SUMO1 substrates, and we found no evidence for specific SUMO1 immunolabelling at either pre- or postsynaptic sites in neurons. Admittedly, based on these findings, we cannot

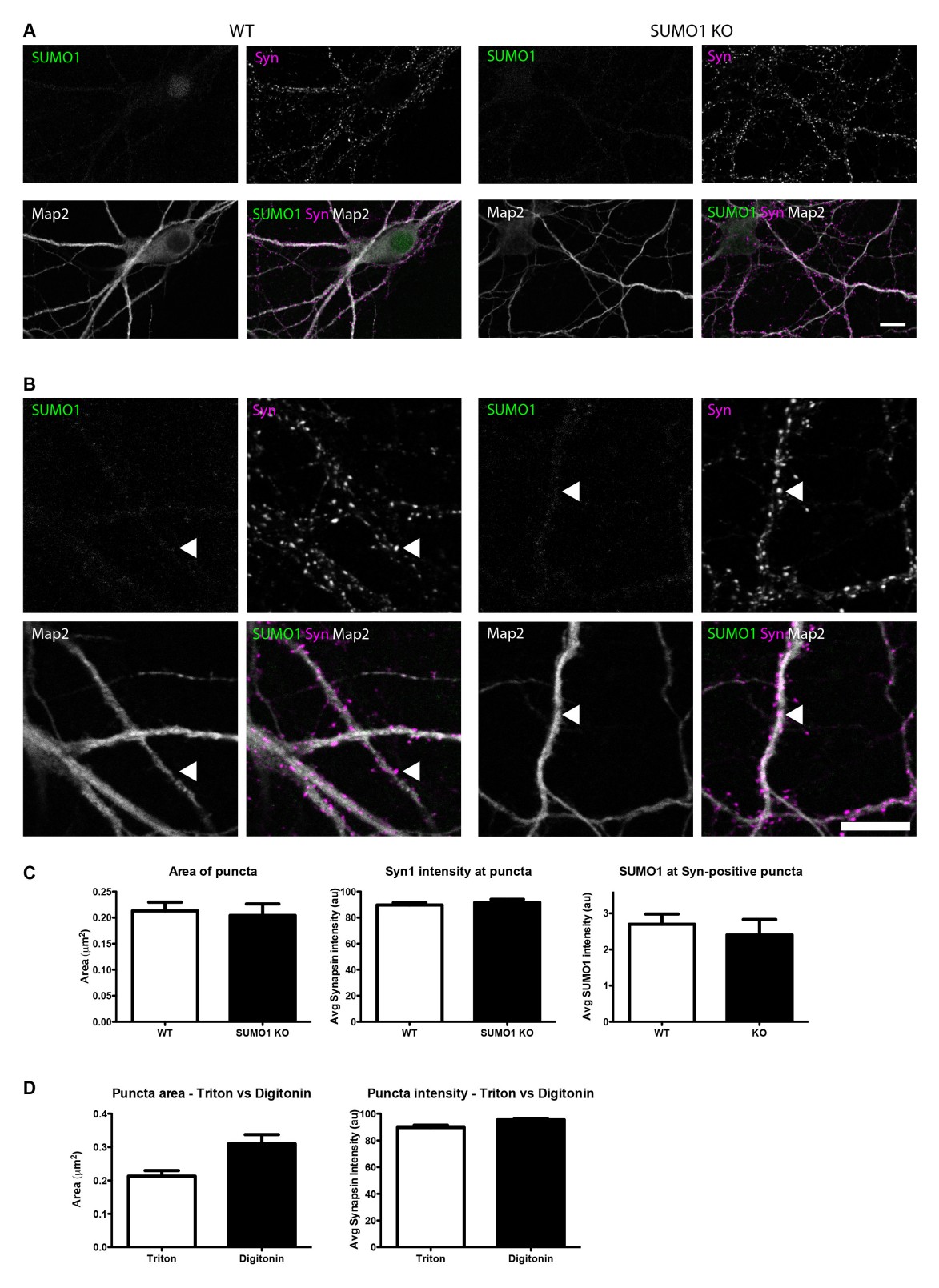

**Figure 13.** Quantification of anti-SUMO1 immunolabelling shows that SUMO1 is not specifically detected at presynaptic sites in Triton X-100-permeabilised neurons. (**A**) Primary hippocampal neurons from WT and SUMO1 KO littermates were fixed, then Triton X-100-permeabilised, and immunolabelled with antibodies against SUMO1 (green), synapsin (purlple), and Map2 (white). A region of interest (neuron) was identified, automatically divided into a grid composed of 6–12 fields of view, and z-stacks were taken for each field. Fields of view were 'stitched' back together to form

*Figure 13 continued on next page*

*Figure 13 continued*

overview images. (**B**) Inset regions from (**A**) are shown in greater detail. (**C**) Synapsin-positive puncta were automatically segmented and the area (left panel) and fluorescence intensity (centre panel) were calculated in synapsin-positive regions of interest (puncta). The intensity of anti-SUMO1 immunolabelling was also quantified at synapsin-positive puncta (right panel). No significant difference between WT and SUMO1 KO neurons was observed for anti-SUMO1 intensity at synapsin-positive puncta. These data indicate that anti-SUMO1 staining at synaptic sites is not specific for SUMO1. (**D**) Graphs comparing the area and intensity of synapsin-positive puncta from WT neurons permeabilised with either Triton X-100 or digitonin. Although the mean area of synapsin puncta was somewhat higher when digitonin was used for permeabilisation, this difference was not quite significant (p=0.057). Images represent immunostaining of cultures from 3 mice of each genotype. Scale bar, 10 μm. Error bars represent SEM.

unequivocally exclude a role of SUMO1-conjugation of synaptic proteins or in synapse function - absence of evidence is not necessarily evidence of absence. On the other hand, our rather stringently controlled findings contrast profoundly with previous reports on SUMO1-conjugation at synapses. In view of this contradictory evidence, it is important to consider why the discrepancies between our conclusions and those of previously published studies have arisen and to discuss strategies that could be employed for the optimal and unequivocal identification and validation of synaptic SUMO substrates.

## Biochemistry

A growing number of synaptic proteins have been described as SUMO1 targets, and consequently, SUMO1-conjugation is considered to be a key regulatory process in synapse function (*Henley et al., 2014*; *Schorova and Martin, 2016*; *Liu et al., 2017*; *Wu et al., 2016*). Strikingly, however, our observations based on Western blotting of fractionated brain tissue indicate very low SUMO1 and SUMO1-conjugate levels in synaptic brain fractions (*Tirard et al., 2012*). It has been argued that our finding in His$_6$-HA-SUMO1 KI mice that SUMO1-conjugates are very rare in synaptic fractions of mouse brain might be due to the slightly decreased (~20% reduction) levels of overall SUMO1-conjugation in the His$_6$-HA-SUMO1 KI mouse brain (*Tirard et al., 2012*; *Henley et al., 2014*; *Luo et al., 2013*). However, we now report that most of the SUMO1 immunoreactive bands that are detected by Western blotting of synaptic fractions, and that have been interpreted as *bona fide* synaptic SUMO1-conjugates, are also observed in samples from KO mice that completely lack SUMO1 (*Figure 6*). This finding illustrates the relevance of using KO or knock-down control samples for the verification of data that are reliant on the performance of antibodies.

However, given the dynamics of protein SUMOylation and the fact that typically only a relatively small fraction of a given protein may be SUMOylated at any time, the lack of detectable SUMO1 in synaptic fractions does not in itself exclude the possibility that synaptic proteins are SUMO1 substrates. Thus, we concentrated additional biochemical analyses on synaptic proteins that had previously been reported as SUMO1 substrates and that represent various synaptic sub-compartments: synapsin 1a, syntaxin 1a, RIM1, gephyrin, GluK2, synaptotagmin1, and mGluR7 (*Matsuzaki et al., 2015*; *Girach et al., 2013*; *Craig et al., 2015*; *Tang et al., 2015*; *Ghosh et al., 2016*; *Choi et al., 2016*; *Martin et al., 2007*). Our main focus was on the SUMO1-conjugation of these candidate proteins in vivo as the main validation criterion. We used anti-HA IP from His$_6$-HA-SUMO1 KI mouse brain to enrich for all SUMO1-modified proteins and then immunoblotted for the proteins of interest. This approach had previously been validated using a number of known SUMO1 substrates (*Tirard et al., 2012*), and was further validated in the present study with the SUMO1 substrate Zbtb20 (*Figure 1*). None of the synaptic proteins examined yielded evidence of SUMO1-conjugation (*Figures 2–5*). Several of the proteins bound non-specifically to affinity beads but showed no specific evidence of the expected molecular weight increase that would be consistent with SUMO1-conjugation. As SUMO1-modified species of synaptic proteins may be of low abundance and hence difficult to enrich when being purified based on the SUMO1 moiety, we reversed the direction of the IP and enriched the proteins of interest with specific antibodies. Again, Western blotting of the purified material with anti-HA antibodies and antibodies against the proteins of interest did not produce evidence for SUMO1-modification of the proteins at hand (i.e. presence of a size-shifted and HA-positive protein species), although the IP had worked robustly. Finally, analyses of SUMO1-conjugation after overexpression of HA-SUMO1 with synapsin1a, GluK2, or gephyrin in HEK293 yielded no evidence of SUMO1-conjugation of these proteins.

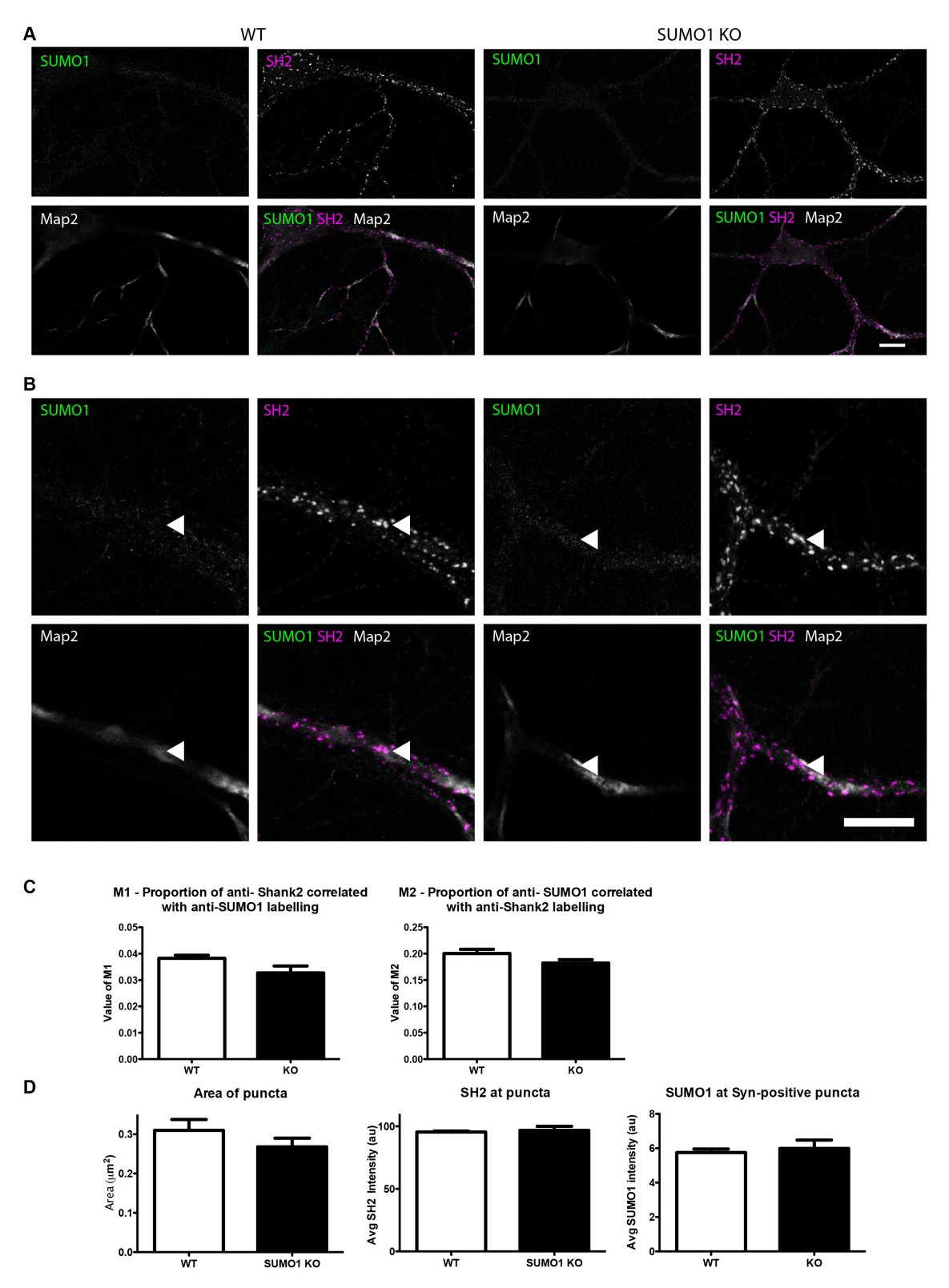

**Figure 14.** Quantification of anti-SUMO1 immunolabelling shows that SUMO1 is not specifically detected at postsynaptic sites in digitonin-permeabilised neurons. (**A**) Primary hippocampal neurons from WT and SUMO1 KO littermates were fixed, then digitonin-permeabilised (10 μg/ml), and immunolabelled with antibodies against SUMO1 (green), shank2 (purple), and Map2 (white). A region of interest (neuron) was identified, automatically divided into a grid composed of 6–12 fields of view, and z-stacks were taken for each field. Fields of view were 'stitched' back together to form

*Figure 14 continued on next page*

Figure 14 continued

overview images. (B) Inset regions from (A) are shown in greater detail. (C) Manders' correlation coefficients between anti-SUMO1 and anti-shank2 were determined. The degree of colocalisation between anti-shank2 and anti-SUMO1 was not different between WT and SUMO1 KO neurons. (D) In a separate analysis, shank2-positive puncta were automatically segmented and the area (left panel) and fluorescence intensity (centre panel) were calculated in shank2-positive regions of interest (puncta). The intensity of anti-SUMO1 immunolabelling was also quantified at shank2-positive puncta (right panel). No significant difference between WT and SUMO1 KO neurons was observed for anti-SUMO1 intensity at shank2-positive puncta. These data indicate that anti-SUMO1 staining at shank2-positive postsynaptic sites is not specific for SUMO1. Images represent immunostaining of cultures from 3 mice of each genotype. Scale bar, 10 µm. Error bars represent SEM.

The discrepancy between our conclusions and those of previous studies may be explained at the biochemical level by considering several key aspects. The first relates to methodology. For a given protein in vivo, its SUMO-conjugated fraction at any given time may only be 1% or less. This makes detection of SUMO substrates highly challenging, especially in vivo. Common approaches to overcome this difficulty are to use systems in which SUMOylation is performed entirely in vitro, using recombinant proteins, or in which the proteins or protein fragments of interest (generally a SUMO isoform plus the proposed SUMO substrate) are overexpressed in a cell line. However, these overexpression systems are not representative of the cellular environment in vivo, so that SUMOylation assessed under those conditions can be the result of off-target artefacts or cell stress rather than the reflection of a physiologically relevant situation. Hence, while overexpression systems are valuable discovery tools in principle, they should not be exclusively relied upon for the validation of a potential SUMO substrate.

The second aspect pertinent to the discrepancy between our observations and previously published studies relates to the criteria based on which a given protein is defined as a SUMO1 substrate. The gold standard for demonstrating that a given protein is SUMOylated relies on Western blotting. Assuming that only one lysine is SUMOylated, the SUMOylated form of the substrate should appear on the blot as a band with a substantially (typically ~20 kDa) higher apparent molecular weight than the unmodified form of the protein. For such a molecular weight shift to be valid as an indicator of SUMOylation, it should be (partly) abolished when the de-SUMOylation enzyme inhibitor NEM is excluded from the buffer in which the cells or tissue were lysed, and should be absent if the target lysine is mutated (ideally to arginine). In the present study, the design of biochemical experiments focused primarily on validating the SUMO1-conjugation of the selected targets in vivo. To this end, we performed complementary IP experiments, that is, either purifying proteins via the conjugated His$_6$-HA-SUMO1 or purifying the candidate itself and then assayed for SUMO1-conjugates by SDS-PAGE and Western blotting for His$_6$-HA-SUMO1 and the candidate target. Importantly, WT mice (for the anti-His$_6$-HA-SUMO1 IP) or purified IgG (for the candidate IP) served as the respective negative controls, which is critical because the use of substantial amounts of antibodies for the IP can generate cross-reactive bands in the final Western blot analysis that can lead to false-positive interpretations. In essence, our data highlight the importance of a combination of well-controlled biochemical approaches in vivo and in vitro as a basis for subsequent functional analyses of candidate protein SUMOylation.

A third issue that requires attention in the present context is the fact that the proposed SUMO1-conjugation of several synaptic proteins (syntaxin1a, synapsin1a, RIM1, mGluR7 and GluK2) is based on work that was carried out in neurons and brains from rats, while the experiments we describe here were carried out exclusively in mice. This raises the possibility that the conflicting findings between previously published reports and our current study are attributable to species-specific differences. However, the sequences between the mouse and rat forms of these proteins are all highly homologous and the lysines proposed to act as sites of SUMO1 conjugation are conserved between mouse and rat. Thus, differences in SUMOylation are very unlikely to be attributable to differences in the protein sequences. In addition, the components of the SUMOylation machinery are conserved between mouse and rat (and indeed across all eukaryotic species), with SUMO1 itself being virtually identical between the two rodent species. In the absence of any reported difference in the levels or machinery of SUMO1-conjugation between rats and mice, it seems unlikely that our observations are simply the result of essential differences between the species.

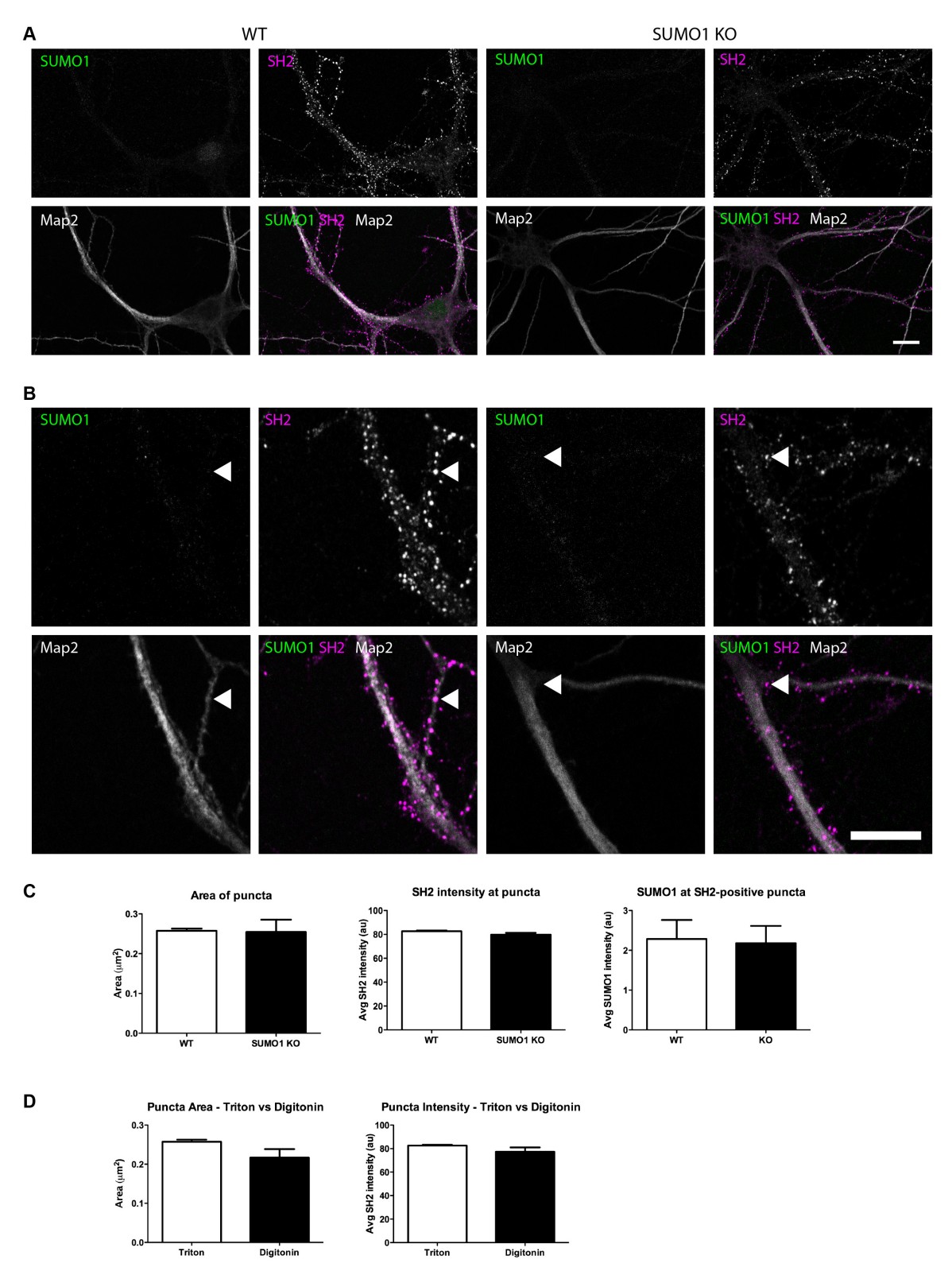

**Figure 15.** Quantification of anti-SUMO1 immunolabelling shows that SUMO1 is not specifically detected at postsynaptic sites in Triton X-100-permeabilised neurons. (**A**) Primary hippocampal neurons from WT and SUMO1 KO littermates were fixed, then triton X-100-permeabilised, and immunolabelled with antibodies against SUMO1 (green), shank2 (purple), and Map2 (white). A region of interest (neuron) was identified, automatically divided into a grid composed of 6–12 fields of view, and z-stacks were taken for each field. Fields of view were 'stitched' back together to form

*Figure 15 continued on next page*

*Figure 15 continued*

overview images. (**B**) Inset regions from (**A**) are shown in greater detail. (**C**) Shank2-positive puncta were automatically segmented and the area (left panel) and fluorescence intensity (centre panel) were calculated in shank2-positive regions of interest (puncta). The intensity of anti-SUMO1 immunolabelling was also quantified at shank2-positive puncta (right panel). No significant difference between WT and SUMO1 KO neurons was observed for anti-SUMO1 intensity at shank2-positive puncta. These data indicate that anti-SUMO1 staining at synaptic sites is not specific for SUMO1. (**D**) Graphs comparing the area and intensity of shank2-positive puncta from WT neurons permeabilised with either Triton X-100 or digitonin. The method of permeabilisation did not change the area or intensity of the shank2-positive puncta. Images represent immunostaining of cultures from 3 mice of each genotype. Scale bar, 10 μm. Error bars represent SEM.

A final issue that requires discussion in this context is the fact that the level of SUMO1-conjugation is slightly reduced in the $His_6$-HA-SUMO1 KI. This might lead to a somewhat reduced sensitivity of SUMO1 detection in this reporter mouse line and hence might have contributed to our inability to detect SUMO1-conjugation of previously proposed SUMO1 substrates. However, our biochemical strategy also employed WT mice with unaltered SUMO1 levels, in which SUMOylated forms of the proposed targets were not detected. Finally, our quantitative immunostaining data on SUMO1 at synapses would also argue against the presence of SUMO1-substrates at synapses.

## Immunostaining of cells and tissues

As a complement to biochemical analyses, recent studies have employed antibodies against SUMO1 in an effort to demonstrate SUMO1 localisation to synapses in cultured neurons. Colocalisation analysis of SUMO1/2/3 and SUMO ligases/peptidases with various synaptic markers were proposed as evidence for synaptic localisation of SUMO1/2/3 (*Jaafari et al., 2013*; *Loriol et al., 2013*, *2012*). Colocalisation between immunolabelling signals for SUMO1 and specific synaptic proteins was also proposed as evidence for the association of SUMO1 with these synaptic proteins (*Girach et al., 2013*; *Martin et al., 2007*; *Konopacki et al., 2011*; *Kantamneni et al., 2011*). These observations are in contrast with two recent studies, in which the localisation of SUMO1 at synapses in vivo was assessed. One immunolabelling study on a mouse line that overexpresses SUMO1 in a neuron-specific manner yielded no evidence for a colocalisation between synaptophysin and SUMO1 (*Matsuzaki et al., 2015*). Further, we observed no localisation of $His_6$-HA-SUMO1 to either excitatory or inhibitory synapses of $His_6$-HA-SUMO1 KI brains, and the same conclusion was drawn from immunolabelling of cultured neurons, where $His_6$-HA-SUMO1 appeared to be localized to nuclei or annulate lamellae, but not to synapses (*Tirard et al., 2012*).

In an effort to reconcile our immunolabelling results with those of previous studies, we took several steps in the current study. We performed immunolabelling against SUMO1, using the well-characterised 21C7 monoclonal antibody in parallel with anti-HA immunolabelling in $His_6$-HA-SUMO1 mice. Neither anti-HA nor anti-SUMO1 immunolabelling indicated specific localisation of SUMO1 at synapses, whereas nuclei exhibited strong immunolabelling with both antibodies. Because a number of studies reported the use of digitonin to permeabilise cells in order to reduce anti-SUMO1 immunolabelling in the nucleus (*Girach et al., 2013*; *Craig et al., 2015*; *Jaafari et al., 2013*), we also compared different permeabilisation methods. Our quantitative analyses of pre- and postsynaptic anti-SUMO1 immunolabelling revealed no difference in anti-SUMO1 labelling between WT and SUMO1 KO neurons. This is in contrast to observations in the nucleus of these neurons, where SUMO1 KO neurons exhibit significantly decreased anti-SUMO1 immunolabelling. These findings indicate that anti-SUMO1 immunolabelling at synapses represents a false-positive signal. Regarding permeabilisation methods and their effects on immunolabelling, the use of Triton X-100 resulted in immunolabelling of neuronal nuclei with either anti-HA or anti-SUMO1 in $His_6$-HA-SUMO1 KI mice, which was greatly reduced in SUMO1 KO neurons. In contrast, digitonin permeabilisation generally prevented nuclear anti-SUMO1 but not anti-HA immunolabelling. That digitonin would specifically abolish anti-SUMO1 immunolabelling in the nucleus but not anti-HA is puzzling. Further, digitonin appears to incompletely permeabilise dendrites and somata, which exhibit 'patchy' anti-Map2 labelling. Thus, at least in our hands, there is some inconsistency with digitonin permeabilisation regarding the access of certain primary antibodies to certain cellular compartments or antigens.

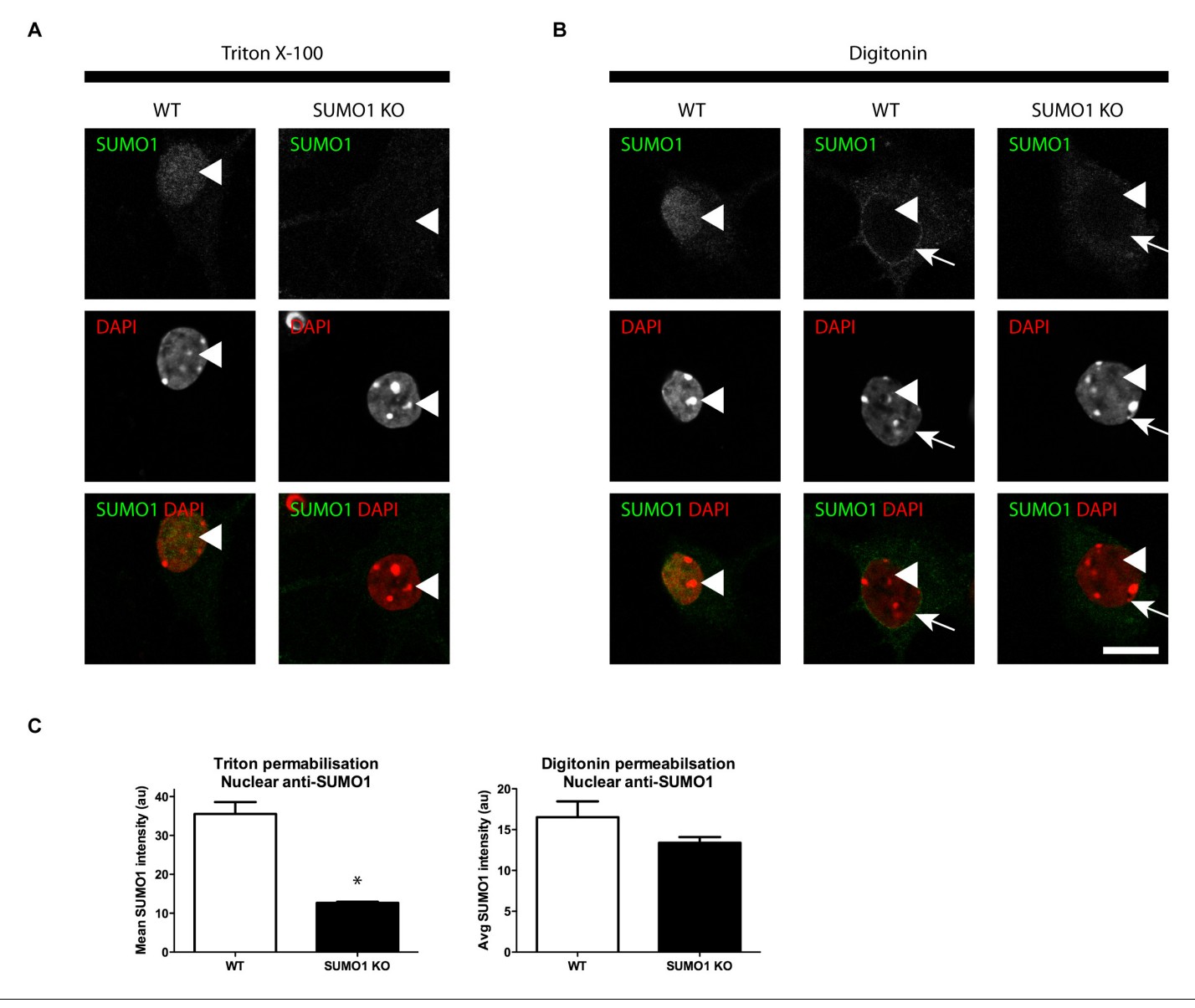

**Figure 16.** Anti-SUMO1 immunolabelling is reduced in the nuclei of SUMO1 KO neurons permeabilised with Triton X-100. Representative immunolabelling against SUMO1 (green) and DAPI (red) in neurons from WT or SUMO1 KO mice is shown. (**A**) In Triton X-100-permeabilised neurons, anti-SUMO1 specifically labelled the nucleus of WT neurons but not of SUMO1 KO neurons (arrowheads). (**B**) In digitonin-permeabilised (10 µg/ml) neurons, neurons occasionally exhibited nuclear anti-SUMO1 immunolabelling (arrowheads, left panels). However, most WT neurons exhibited no nuclear anti-SUMO1 immunolabelling when permeabilised with digitonin (arrowheads, centre panel), though a ring of anti-SUMO1 immunolabelling was often evident around the nucleus (arrows). SUMO1 KO neurons permeabilised with digitonin exhibited no anti-SUMO1 immunolabelling within or around the nucleus (right panels, arrows and arrowheads). (**C**) Graphs showing the average intensity of anti-SUMO1 immunolabelling within the nucleus of Triton X-100- and digitonin-permeabilised neurons. Overall, SUMO1 KO neurons exhibited a significant reduction in nuclear anti-SUMO1 immunolabelling intensity when permeabilised with Triton X-100 but not with digitonin. Images represent immunostaining of cultures from 3 mice of each genotype. Scale bar, 10 µm. Error bars represent SEM.

Limited validation of anti-SUMO1 immunolabelling in neurons may partially explain the widely varying localisation patterns of anti-SUMO1 immunolabelling reported in previous studies. In dendrites, anti-SUMO1 immunolabelling has appeared as highly punctate in some studies (*Craig et al., 2015*; *Martin et al., 2007*; *Konopacki et al., 2011*; *Loriol et al., 2013*) and relatively homogenous in others (*Ghosh et al., 2016*; *Craig et al., 2012*; *Kantamneni et al., 2011*). Several

studies have also shown seemingly paradoxical changes in immunostaining intensity, with glycine-induced chemical LTP (*Jaafari et al., 2013*; *Kantamneni et al., 2011*), stimulation of neuronal activity (*Craig et al., 2015*), and inhibition of neuronal activity (*Craig et al., 2012*) all causing increased anti-SUMO1 immunolabelling in dendrites. Repetition of these treatments with SUMO1 knockout or knock-down controls may clarify this matter. Finally, with regards to studying specific SUMO1 substrates using immunolabelling, a correlation-based analysis may be of limited use as colocalisation is not reflective of the SUMO1-conjugation state of the proposed substrate protein, unless it is demonstrated that colocalisation is lost or diminished when the target lysines of the given substrate are mutated.

In sum, our present findings provide no evidence of the presence of SUMO1 at synapses. This raises concerns regarding the use of anti-SUMO1 immunolabelling-based approaches in the absence of a SUMO1 KO or knock-down control. Further, it seems important that negative controls for antibody labeling of SUMO1 are provided whenever immunolabelling conditions are changed, since altering permeabilisation conditions, for example, can have marked effects.

### Consensus criteria for studies on SUMOylation in synapse biology

While SUMO1-conjugation of transcription factors can have profound - but indirect - effects on synaptic function (*Tai et al., 2016*; *Estruch et al., 2016*; *Shalizi et al., 2006*), our present data indicate, contrary to previous studies, that the role of SUMO1-modifications in the function of synaptic proteins and synapses remains unclear. To ultimately resolve this issue and the related controversy in the field, it will be necessary to define a consensus regarding the criteria that have to be met so that a candidate protein can be considered a *bona fide* SUMO substrate. Although not concerned with SUMOylation in neurons, landmark studies by pioneers in the field of SUMO biology can serve as 'blueprints' in this context (*Barysch et al., 2014*; *Tatham et al., 2009*; *Xiao et al., 2015*; *Knipscheer et al., 2009*). Translated to nerve cell biology, the corresponding criteria can be summarized as follows:

1. Immunoprecipitation of SUMO-conjugated proteins from brain followed by Western blotting against the candidate protein and the respective SUMO isoform must show that the protein of interest is present in the immunopurified sample and exhibits a molecular weight increase that is consistent with SUMO-conjugation.
2. Immunoprecipitation from the brain using an antibody against the candidate protein followed by Western blotting against the candidate and the respective SUMO isoform must show that the protein of interest is enriched in the immunopurified sample and that a variant of the candidate protein exhibits a molecular weight increase that is consistent with SUMO-conjugation.
3. The size shift that reflects a SUMO-conjugated protein variant in Western blotting experiments should be (partially) sensitive to the omission of the de-SUMOylation enzyme inhibitor NEM. Mutation of SUMO-acceptor lysines must result in a (partial) loss of SUMOylation as assessed with the methods described above.
4. In case that immunostaining is used to support SUMOylation of a candidate protein, the specificity of the antibodies against SUMO and the selected candidate must be established by the loss of signal upon KO or knock-down. This control is also advisable for Western blot approaches.

It is likely that the application of these criteria, along with stringent controls for antibody specificity (i.e. validation using KO of knock-down samples), will aid substantially in determining the exact neuronal and synaptic function of SUMO1-conjugation in particular and of SUMOylation in general.

## Materials and methods

### Mice

The His$_6$-HA-SUMO1 knock-in (RRID:MGI:5476976), the SUMO1 knock-out (RRID:MGI:3810737) and the Zbtb20 knock-out (RRID:MGI:5693950) lines were described previously (*Tirard et al., 2012*; *Zhang et al., 2008*; *Rosenthal et al., 2012*). All animal experiments were performed in accordance with the guidelines for the welfare of experimental animals issued by the State Government of Lower Saxony, Germany, in compliance with European and NIH guidelines (33.9-42502-04-13/1359).

## Cell culture and transfection

The GFP-gephyrin construct was previously described (*Fuhrmann et al., 2002*), the HA-SUMOs construct were kind gift from Dr. Frauke Melchior (Heidelberg, Germany), GFP-synapsin1a overexpression construct was a kind gift from Dr. D. Gitler (Beer-Sheva, Israel), Gluk2 was a kind gift of Drs. C. Mulle and F. Coussen, Zbtb20 overexpression construct was a kind gift of Dr. Peter Penzes (Chicago, USA). HEK293-FT cells (Gibco, RRID:CVCL_6911) were grown at 37°C with 4% CO2 and maintained in DMEM (Gibco, Life Technologies, Darmstadt, Germany), 10% (v/v) fetal calf serum (Gibco, Life Thechnologies, Darmstadt, Germany), 50 units/ml penicillin and 50 units/ml streptomycin (Roche Applied Science, Mannheim, Germany). Transfection was carried out using Lipofectamine 2000 (Invitrogen, Karlsruhe, Germany) following the manufacturer's protocol. Cells were assumed to be authenticated by the supplier and were not further confirmed in this study. Mycoplasma contamination was monitored by DAPI staining.

The cell line used in this study does not belong to the list of misidentified cell lines defined by the International Cell Line Authentication Committee.

## Preparation of total brain homogenate

Brains of embryonic day 18 animals were homogenized in 10 ml of lysis buffer (150 mM NaCl, 1% Triton X-100, 10 mM Tris, pH 7.6) containing protease inhibitors (1 µg/ml aprotinin, 0.5 µg/ml leupeptine, 17.4 µg/ml PMSF) using a glass-Teflon homogenizer (900 rpm, 12 strokes). The brain homogenate was centrifuged at 1000 x *g* for 10 min at 4°C to remove debris. The protein concentration in the resulting supernatant was determined using the BCA method. Samples were flash frozen in liquid nitrogen and kept at −80°C until being processed for SDS-PAGE and Western blot analysis.

## Subcellular fractionation of brain tissue

Subcellular fractions were prepared essentially as described previously (*Tirard et al., 2012*; *Jones and Matus, 1974*). Brains were homogenized in 10 ml of 320 mM sucrose containing 4 mM HEPES pH7.4, protease inhibitors (1 µg/ml aprotinin, 0.5 µg/ml leupeptine, 17.4 µg/ml PMSF), and 20 mM NEM using a glass-Teflon homogenizer (900 rpm, 12 strokes). The brain homogenate (H) was centrifuged at 1000 x *g* for 10 min at 4°C in an SS-34 rotor (Sorvall, Thermo Fischer Scientific, Langenselbold, Germany). The supernatant (S1) was removed from the pellet (P1) and centrifuged at 12,500 x *g* for 15 min at 4°C in an SS-34 rotor. The supernatant (S2) was removed completely and the synaptosome-enriched pellet (P2) was resuspended in 9 volumes of cold water containing 4 mM HEPES pH7.4 and homogenized using a glass-Teflon homogenizer (1500 rpm, 10 strokes). The homogenized sample was then centrifuged for 20 min at 4°C in an SS-34 rotor at 25,000 x g. The resulting supernatant (LS1) was ultracentrifuged at 200,000 *g* for 2 hr at 4°C to generate fractions LP2 and LS2. The pellet (LP1) was resuspended in 1 ml of homogenization buffer and layered on top of a two-step sucrose density gradient (5 ml of 1.2 M and 5 ml of 0.8 M sucrose, 4 mM HEPES, protease inhibitors as above). The gradient was centrifuged at 62,000 x g for 120 min at 4°C in an SW-41Ti rotor (Beckman, Krefeld, Germany). Synaptic membranes were recovered at the interface of 0.8 M and 1.2 M sucrose using a Pasteur pipette. The resulting fraction was then pelleted using an SS-34 rotor at 4°C for 20 min at 37,000 x g and is referred to as SPM fraction. The various brain fractions are designated as follows: H, homogenate; P1, nuclear pellet; S1, supernatant after P1 sedimentation; P2, crude synaptosomal pellet; S2, supernatant after P2 sedimentation; LP1, lysed synaptosomal membranes; LS1, supernatant after LP1 sedimentation; SPM, synaptic plasma membranes.

## Immunoprecipitations

Anti-HA affinity purification has been described previously (*Tirard et al., 2012*; *Tirard and Brose, 2016*). For immunoprecipitation of specific proteins (syntaxin1a, gephyrin, GluK2, synapsin1, RIM1) from whole brain or from HEK cells, tissue or cells were homogenized in lysis buffer (150 mM NaCl, 1% Triton X-100, 10 mM Tris, pH 7.6) containing protease inhibitors (1 µg/ml aprotinin, 0.5 µg/ml leupeptine, 17.4 µg/ml PMSF) and 20 mM NEM, sonicated, and ultracentrifuged at 100,000 x *g* for 1 hr at 4°C. The resulting supernatant was preincubated with either Protein A or Protein G Sepharose beads (GE Healthcare, Feibrug, Germany) and antibodies (specific IgG or control IgG from the same species) or anti-HA beads (Sigma-Aldrich, Taufkirchen, germany) were added to the supernatant.

After incubation for 4 hr at 4°C on a rotating wheel, the beads were pelleted and washed repeatedly in lysis buffer. Bound material was eluted directly into SDS-PAGE sample buffer.

## Immunoblotting

SDS-PAGE was performed with standard discontinuous gels (*Laemmli, 1970*) or with commercially available 4–12% Bis-Tris gradient gels (Invitrogen). Western blots were probed using primary and secondary antibodies as indicated below. Blots were routinely developed using enhanced chemiluminescence (GE Healthcare).

## Primary neuron culture and immunocytochemistry

Hippocampal neurons were prepared as described (*Bekkers and Stevens, 1991*; *Rosenmund and Stevens, 1996*). Primary neuron cultures were fixed for 20 min in cold PB containing 4% PFA, pH 7.4. Coverslips were then washed repeatedly in PBS. Blocking and permeabilisation were performed for 20 min at room temperature in PBS containing 3% fetal bovine serum (FBS) and 0.3% Triton X-100 or 10 µg/ml digitonin or 20 µg/ml of digitonin. Primary antibodies were applied for 2 hr at room temperature in PBS containing 3% FBS. Coverslips were then washed repeatedly with PBS and incubated with appropriate secondary antibodies (conjugated to AlexaFluor 488, 555, or 633) for 1 hr at room temperature in PBS containing 3% FBS. Coverslips were then stained with DAPI (in PBS), washed twice in PBS and mounted on slides with Aquapolymount (Polysciences, Hirschberg an der Bergstrasse, Germany). The antibodies used are listed below. For the quantitative imaging comparing WT and SUMO1 KO neurons, neuron cultures were prepared from three separate litters of pups, with one WT and one SUMO1 KO animal from each litter. Cells were at 16–17 DIV at the time of fixation and immunolabelling. To minimise inter-sample variability in immunolabelling, fixation and immunolabelling were performed on all samples in a single batch, such that the same reagents were used for all samples. All images for a given condition were taken from a single coverslip.

## Antibodies

Primary antibodies used in IPs, Western blotting (WB) and immunocytochemistry (ICC) were as follows: mouse anti-SUMO1 21C7 (Hybridoma Bank, Iowa, WB: 1/1000, ICC: 1/50, RRID:AB_2198257), mouse anti-HA (Biolegend, 901515, WB: 1/1000, ICC: 1/100, RRID:AB_2565334) mouse anti-synapsin1 (Synaptic Systems, 106021, WB: 1/1000, RRID:AB_2617072), mouse anti-syntaxin1a (Synaptic Systems, 110111, WB: 1/1000, RRID:AB_887848), mouse anti-synaptotagmin (Synaptic System, 105001, WB: 1/1000, RRID:AB_887831), mouse anti-gephyrin (Synaptic Systems, 147111, WB: 1/1000, RRID:AB_887719), rabbit anti-RIM1 (Synapsin Systems, 140003, WB: 1/1000, RRID:AB_887774), rabbit anti-GluK2 (Millipore, 04–921, WB: 1/1000, RRID:AB_1587072), rabbit anti-mGlur7 (Upstate, 07–239, WB: 1/1000, RRID:AB_310459), mouse anti-GluN1 (Synaptic Systems, 114011, WB: 1/1000, RRID:AB_887750), mouse anti-synaptophysin (Synaptic Systems, 101011, WB: 1/1000, RRID:AB_887824), rabbit anti-synapsin1/2 (Synaptic Systems, 106002, ICC: 1/2000, RRID:AB_887804), rabbit anti-shank2 (Synaptic Systems, 162202, ICC: 1/1000, RRID:AB_2619860), chicken anti-Map2 (Novus, NB300213, ICC: 1/1000, RRID:AB_2138178), mouse anti-actin (Sigma-Aldrich, clone AC-40, WB: 1/5000, RRID:AB_476730).

Secondaries antibodies were as follows: HRP conjugated goat anti-mouse (Biorad, 172–1011, WB: 1/5000, RRID:AB_11125936), HRP conjugated goat anti-rabbit (Biorad, 172–1019, WB: 1/5000, RRID:AB_11125143), goat anti-chicken Alexa Fluo-633 (Thermo Fischer, A-11039, ICC: 1/1000, RRID: AB_2534096), goat anti-mouse Alexa Fluo-488 (Life, A11029, ICC: 1/1000), goat anti-rabbit Alexa Fluo-555 (Thermo Fischer, A-21429, ICC: 1/1000, RRID:AB_2535850).

## Imaging and image analysis

All imagings were performed using a Leica SP5X Confocal microscope and a 100x objective (NA = 1.4). For quantitative imaging of anti-SUMO1 immunolabelling, a white-light laser (lines 488, 555, 633 nm) was used in combination with a resonant scanner and Leica HyD hybrid detectors to maximise image acquisition speed and maximise image signal-to-noise ratio, respectively. DAPI was imaged using a 405 nm laser. Images were acquired at 12-bit. All microscope settings (laser power, detector settings) were kept the same between samples and imaging sessions. During a single imaging session, images were acquired from one WT or SUMO1 KI sample and the corresponding

littermate SUMO1 KO sample. Neurons were selected for imaging on the basis of immunolabelling for synapsin or shank2, and anti-SUMO1 immunolabelling was not visualised through the eyepiece prior to image acquisition. This was to ensure that the anti-SUMO1 imaging was performed in an unbiased manner. Once a region of interest was identified (usually a neuronal soma and primary dendrites), the zoom was set to 5x within the software (giving 60.7 nm/pixel), and a tiling function was used to divide this region into 6–12 slightly overlapping fields of view, which would then be imaged sequentially. Within each field of view, a z-stack of images was acquired to ensure that the complete thickness of the neuron was imaged. These images could then be 'stitched' together to give a single high-resolution image stack encompassing a large region of the neuron. For analysis of anti-SUMO1, a z-plane was chosen that was optimal for either synapses/dendrites (for synaptic quantification) or the nucleus (for nuclear quantification). Images from that plane were then assembled into a single image using the Grid/Pairwise Stitching plugin in Fiji (*Preibisch et al., 2009*). These large stitched images were then subjected to quantitative analysis. For nuclear anti-SUMO1 labelling intensity analysis, images of anti-SUMO1 and DAPI labelling were converted to eight-bit, the DAPI image was automatically thresholded (Li's Minimum Cross Entropy Thresholding method), and the thresholded image was used to create a 'mask' representing the cell nucleus. The average anti-SUMO1 signal within the region defined by the mask was then calculated. For correlation analyses, images showing anti-SUMO1 and either anti-synapsin or anti-shank2 labelling were analysed using the JACoP plugin for ImageJ (*Bolte and Cordelières, 2006*). Manders M1 and M2 coefficients were calculated after automatic thresholding within the plugin. The M1 and M2 were then compared between WT and SUMO1 KO neurons. For quantification of the anti-SUMO1 immunolabelling localised to either anti-synapsin or anti-shank2 puncta, a custom-written macro for Fiji was used. Briefly, the macro defined individual anti-synapsin or anti-shank2 puncta as regions of interest (ROIs), and then calculated the intensity of anti-SUMO1 immunolabelling within these ROIs. Average anti-SUMO1 labelling intensity within the ROIs was then compared between WT and SUMO1 KO neurons.

## Statistical analysis and data collection

All statistical tests were performed using Graphpad Prism. The level of statistical significance was set at $p < 0.05$ for all tests. For comparisons of two sample means, Student's *t*-test was used, while for comparing more than two sample means one-way ANOVA was used with *post-hoc* Bonferroni. For the purposes of quantitative immunolabelling, images were taken of 4–6 neurons per coverslip. Statistical independence was defined at the level of individual animals, rather than individual synaptic puncta or neurons (*Galbraith et al., 2010*), that is, data from individual synaptic puncta and/or neurons were averaged to give a single value for each animal.

## Acknowledgements

We thank K Hellmann for excellent technical support, F Benseler, I Thanhäuser, D Schwerdtfeger, and C Harenberg for oligonucleotide synthesis and DNA sequencing, L van Werven for peptide synthesis, and the staff of the Max Planck Institute of Experimental Medicine Transgenic Animal Facility for mouse husbandry.

## Additional information

### Funding

| Funder | Grant reference number | Author |
| --- | --- | --- |
| Max-Planck-Gesellschaft | Open-access funding | Marilyn Tirard |

The funders had no role in study design, data collection and interpretation, or the decision to submit the work for publication.

### Author contributions

JAD, Conceptualization, Formal analysis, Supervision, Validation, Investigation, Methodology, Writing—original draft, Writing—review and editing; BHC, Software, Visualization, Methodology; JJP, F-PZ, Resources, Methodology, Writing—review and editing; NB, Conceptualization, Supervision,

Funding acquisition, Validation, Investigation, Methodology, Writing—original draft, Writing—review and editing; MT, Conceptualization, Formal analysis, Supervision, Investigation, Methodology, Writing—original draft, Writing—review and editing

## Author ORCIDs

James A Daniel, http://orcid.org/0000-0002-2781-4544
Marilyn Tirard, http://orcid.org/0000-0002-5669-9610

## Ethics

Animal experimentation: All animal experiments were performed in accordance with the guidelines for the welfare of experimental animals issued by the State Government of Lower Saxony, Germany, in compliance with European and NIH guidelines (33.9-42502-04-13/1359).

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
