## [Decision Letter]

Thank you for submitting your article "Analysis of SUMO1-conjugation at Synapses" for consideration by *eLife*. Your article has been reviewed by two peer reviewers, and the evaluation has been overseen by Gary Westbrook as the Senior and Reviewing Editor. The reviewers though that this was an important topic and worthy of publication in *eLife* if the comments can be addressed. The reviewers have opted to remain anonymous.

The reviewers have discussed the reviews with one another there was a agreement on the substance of both reviews in particular the issue raised by reviewer 2.

Reviewer #1:

As a major form of posttranslational modification, SUMOylation has been considered playing significant roles in various aspects of cellular functions. Evidence from recent studies, with the identification of a number of SUMOylated synaptic proteins, suggests that SUMOylation also plays important roles in regulating synaptic functions. However, stringent validation of SUMOylation of endogenous synaptic proteins is still lacking. The work by Daniel et al., uses a knock-in reporter mouse line expressing tagged SUMO1 and a SUMO1 knockout mouse to investigate SUMO1 conjugation at synapses. The authors failed to find evidence supporting SUMO1-conjugation of seven synaptic proteins that previously reported to be SUMOylated. Additionally, it was suggested that previously reported SUMO1 immunoreactivity at synapses may be non-specific. Instead, SUMO1 conjugation is mainly in the nuclear fraction and SUMO1 immunoreactivity is detected mainly in the nucleus. Thus, the current study calls for a 'pause' in our consideration of roles of SUMO1 conjugation at synapses, and asks the field to step back and reevaluate evidence for synaptic SUMOylation with a series of stringent criteria and controls. This study is a true tour de force in the investigation of SUMO1-conjugation at synapses. It not only is highly significant in its experimental outcome – an example that negative results can be very important, but also sets the standard for studying posttranslational modification of postsynaptic proteins. I do not have any additional experimental requests because the study is very thorough and convincing to me.

*Reviewer #2:*

This manuscript determines the abundance of sumoylation of synaptic proteins. It re-assesses a body of work (published in many papers, some in high impact journals) that suggested that several pre- and postsynaptic proteins are sumoylated, and that sumoylation provides for important regulatory mechanisms. It employs a previously generated knockin mouse line in which SUMO1 is tagged with HA, and then assesses in these mice, in wild-type mice, and in SUMO1 knockout mice whether sumoylation of synaptic proteins can be detected. It pursues this question rigorously using well-controlled biochemical experiments and morphological analyses. The authors find no evidence that any of the tested proteins (synapsin, syntaxin-1, RIM, GluK2, gephyrin, mGluR7 and synaptotagminh-1) are sumoylated, and it does not find any evidence for sumoylation at synapses using morphology. It concludes that many of the previous studies may have been mislead by non-rigorous assessment of sumoylation, and proposes general guidelines that should be followed to assess whether a synaptic protein is sumoylated.

The data are carefully acquired and the manuscript is well written. It also takes a balanced approach in discussing previous data even though it does not confirm these data. Although it essentially reports negative data, I consider this an important body of work, as it addresses whether synaptic proteins are sumoylated more rigorously and convincingly than previous manuscripts that came to different conclusions. I am generally in support of publishing this manuscript in *eLife*. However, the following point should be considered before publication:

Experiment:

A key issue that has been raised before is whether sumoylation is impaired by the HA-SUMO1 knockin mutation. To quantitatively compare how SUMO1-conjugation is impaired in these mice, the authors should compare SUMO-1 antibody labeling intensity in neuronal nuclei using immunofluorescence in HA-SUM0-1 mice compared to wt mice (next to the comparison of WT and SUMO-KO mice in Figure 14C). This would add confidence that in sumoylation can be detected in these neurons, would experimentally disclose that sumoylation is impaired, and would provide a gage for how much it is impaired. A related, qualitative experiment is shown in Figure 8A, but nuclear SUMO1 labeling in that figure is not convincing.

In addition, the authors argue in subsection “Anti-SUMO1 immunolabelling shows no SUMO1 at synapses in His6-HA-SUMO1 KI neurons” (in the beginning of the paragraph following the subheading) that: " It was argued previously that the His6 -HA tag added to SUMO1 in the His6 -HA-SUMO1 KI might prevent SUMO1 from modifying key substrates (Luo et al., 2013), although the SUMO1-conjugation pattern as assessed by SDS-PAGE and Western blotting appeared to be very similar in His6 -HA-SUMO1 KI and control samples (Figure 6), excluding major defects […]". The only control in Figure 6 is a SUMO1 KO, it is hard to follow the logic of the argument.

---

## [Author Response]

Reviewer #1:

*As a major form of posttranslational modification, SUMOylation has been considered playing significant roles in various aspects of cellular functions. Evidence from recent studies, with the identification of a number of SUMOylated synaptic proteins, suggests that SUMOylation also plays important roles in regulating synaptic functions. However, stringent validation of SUMOylation of endogenous synaptic proteins is still lacking. The work by Daniel et al., uses a knock-in reporter mouse line expressing tagged SUMO1 and a SUMO1 knockout mouse to investigate SUMO1 conjugation at synapses. The authors failed to find evidence supporting SUMO1-conjugation of seven synaptic proteins that previously reported to be SUMOylated. Additionally, it was suggested that previously reported SUMO1 immunoreactivity at synapses may be non-specific. Instead, SUMO1 conjugation is mainly in the nuclear fraction and SUMO1 immunoreactivity is detected mainly in the nucleus. Thus, the current study calls for a 'pause' in our consideration of roles of SUMO1 conjugation at synapses, and asks the field to step back and reevaluate evidence for synaptic SUMOylation with a series of stringent criteria and controls. This study is a true tour de force in the investigation of SUMO1-conjugation at synapses. It not only is highly significant in its experimental outcome – an example that negative results can be very important, but also sets the standard for studying posttranslational modification of postsynaptic proteins. I do not have any additional experimental requests because the study is very thorough and convincing to me.*

We have now added a more detailed description and explanation of the size-shift caused by SUMO1-conjugation of Zbtb20 in cultured cells and in brain. This information has been added to the text that deals with what has become Figure 1 in the revised manuscript (Results, subsection “Zbtb20 as an example of reliable validation of a SUMO1 substrate in vivo and in vitro using the His6-HA-SUMO1 KI”). In addition, we added Figure 1—figure supplement 2, which describes a further validation of the anti-Zbtb20 antibody we used and we inserted a panel of a shorter exposure of the Western blot of the Zbtb20 IP (Figure 1, two top panels on the right) that we believe illustrates more appropriately the various Zbtb20 forms that are enriched during the IP.

As a general comment, the anti-HA IP (Figure 1) is more efficient than the anti-Zbtb20 IP (Figure 1) in enriching SUMOylated Zbtb20. SUMO1 conjugated Zbtb20 is weakly enriched by the anti-Zbtb20 IP. Moreover, because we are using the same antibody to perform the anti-Zbtb20 IP and the Western blot testing, cross-reactive bands representing IgG are detected in the eluate fraction, which renders the detection of SUMO1-conjugated Zbtb20 more difficult. This issue is briefly discussed (Discussion, subsection “Biochemistry” paragraph four).

Further, we performed SDS-PAGE using a 4-12% acrylamide gel for Figure 1, an 8% gel for Figure 1, and a 10% gel for Figure 1. This is now explained in the corresponding legend. Varying the type of gel can influence the size-shift on the gel that is induced by SUMO1-conjugation. This now briefly mentioned in the text (subsection “Design of biochemical experiments”). Further, pre-stained protein markers differ somewhat from batch to batch, which introduces some variability in protein size assessment.

Despite the issues addressed above, Zbtb20 shows a clear shift in size under the various experimental conditions, so that our corresponding data are a clear indication of the SUMO1-conjugation of Zbtb20 in vivo and in vitro.

*Reviewer #2:*

*This manuscript determines the abundance of sumoylation of synaptic proteins. It re-assesses a body of work (published in many papers, some in high impact journals) that suggested that several pre- and postsynaptic proteins are sumoylated, and that sumoylation provides for important regulatory mechanisms. It employs a previously generated knockin mouse line in which SUMO1 is tagged with HA, and then assesses in these mice, in wild-type mice, and in SUMO1 knockout mice whether sumoylation of synaptic proteins can be detected. It pursues this question rigorously using well-controlled biochemical experiments and morphological analyses. The authors find no evidence that any of the tested proteins (synapsin, syntaxin-1, RIM, GluK2, gephyrin, mGluR7 and synaptotagminh-1) are sumoylated, and it does not find any evidence for sumoylation at synapses using morphology. It concludes that many of the previous studies may have been mislead by non-rigorous assessment of sumoylation, and proposes general guidelines that should be followed to assess whether a synaptic protein is sumoylated.*

*The data are carefully acquired and the manuscript is well written. It also takes a balanced approach in discussing previous data even though it does not confirm these data. Although it essentially reports negative data, I consider this an important body of work, as it addresses whether synaptic proteins are sumoylated more rigorously and convincingly than previous manuscripts that came to different conclusions. I am generally in support of publishing this manuscript in eLife. However, the following point should be considered before publication:*

*Experiment:*

*A key issue that has been raised before is whether sumoylation is impaired by the HA-SUMO1 knockin mutation. To quantitatively compare how SUMO1-conjugation is impaired in these mice, the authors should compare SUMO-1 antibody labeling intensity in neuronal nuclei using immunofluorescence in HA-SUM0-1 mice compared to wt mice (next to the comparison of WT and SUMO-KO mice in Figure 14C). This would add confidence that in sumoylation can be detected in these neurons, would experimentally disclose that sumoylation is impaired, and would provide a gage for how much it is impaired. A related, qualitative experiment is shown in Figure 8A, but nuclear SUMO1 labeling in that figure is not convincing.*

We agree with this reviewer in that it is important to understand how SUMO1 conjugation in the His_6_-HA-SUMO1 KI mouse compares to the characteristics of WT mice. In this regard, we note that our previous study (Tirard et al., 2012) documented by Western blot analysis of brain tissue that SUMO1 conjugation to proteins is reduced by ~20% in His_6_-HA-SUMO1 KI mice as compared to WT mice. We also note that in this previous study a number of known SUMO1 substrates were verified as SUMO1 substrates in His_6_-HA-SUMO1 KI mouse brain. As such, we feel that the key point that this reviewer raises, which is whether SUMO1 conjugation in KI mice can be detected and the extent to which it is reduced compared to WT mice, has already been addressed in vivo. This aspect is now described and discussed more explicitly and clearly in the revised manuscript (Results, subsection “The His6-HA-SUMO1 Knock-In Mouse Model” and subsection “Anti-SUMO1 immunolabelling shows no SUMO1 at synapses in His6-HA-SUMO1 KI neurons” and in the Discussion section).

We have further taken the view that this comment may relate to evaluating how this ~20% reduction in the level of SUMO1 conjugates observed by Western blot in vivo translates to (nuclear) anti-SUMO1 immunolabelling in vitro. We performed corresponding experiments and added the data to the revised manuscript as Figure 1 to 8—figure supplement 1, along with a description in main text (Results, subsection “The His6-HA-SUMO1 Knock-In Mouse Model”).

All quantitative immunolabelling data in the original manuscript were generated by comparing cultures from littermate WT and SUMO1 KO mice. All fixation, immunollabelling, and imaging analyses were then performed in parallel, resulting in a dataset with minimal experimental variability. At present we do not have littermate WT and His_6_-HA-SUMO1 KI mice as a homozygous His_6_-HA-SUMO1 KI line and a parallel WT line derived from the same founders have been used so far to facilitate large-scale biochemistry experiments. Generating WT and His_6_-HA-SUMO1 KI mice in the same litter would require additional rounds of mouse breeding which would take several months. Thus, in the interest of responding within a reasonable timeframe to *eLife*, we performed a comparison of nuclear anti-SUMO1 immunolabelling between WT cultures and His_6_-HA-SUMO1 KI obtained from the parallel lines as detailed below.

In performing this experiment, we compared mice from two litters of WT mice (fixed at 15 and 12 days in vitro) with one litter of KI mice (fixed at 11 days in vitro). All cells were fixed and immunolabelled at the same time as this is essential to minimize inter-sample variability in immunolabelling. However, SUMO1-conjugation levels increase with brain development from postnatal day 0 to postnatal day 15 (Tirard et al., 2012, Figure S2), a phenomenon that was also apparent after comparing nuclear SUMO1 labeling of neurons at 12 and 15 days in vitro. To compare roughly age-matched datasets, we therefore only analyzed two WT cultures at 12 days in vitro and three His_6_-HA-SUMO1 KI cultures at 11 days in vitro. The corresponding data, which are now shown in Figure 1 to 8—figure supplement 1 and discussed in the main text (Results, subsection “The His6-HA-SUMO1 Knock-In Mouse Model” and subsection “Anti-SUMO1 immunolabelling shows no SUMO1 at synapses in His6-HA-SUMO1 KI neurons” and Discussion section) indicate a reduction of ~35% in the mean nuclear anti-SUMO1 immunolabelling in His_6_-HA-SUMO1 KI neurons, which is likely an overestimation because the WT cells were one day older. Despite this reduction, the images and quantification demonstrate that SUMO1 conjugation can be detected in the His_6_-HA-SUMO1 KI mice by immunolabelling.

Further, we mention at different places in the manuscript that we cannot rule out the possibility that this decrease in SUMO1 levels in the His_6_-HA-SUMO1 KI may somewhat limit our ability to detect SUMO1 (Results, subsection “The His6-HA-SUMO1 Knock-In Mouse Model” and subsection “Anti-SUMO1 immunolabelling shows no SUMO1 at synapses in His6-HA-SUMO1 KI neurons” and Discussion section). On the other hand – and importantly – our quantitative analysis of anti-SUMO1 immunolabelling was performed using WT and SUMO1 KO neurons and shows that we do not detect specific synaptic anti-SUMO1 immunolabelling in WT cultures (Figures 9 to 12).

*In addition, the authors argue in subsection “Anti-SUMO1 immunolabelling shows no SUMO1 at synapses in His6-HA-SUMO1 KI neurons” (in the beginning of the paragraph following the subheading) that: " It was argued previously that the His6 -HA tag added to SUMO1 in the His6 -HA-SUMO1 KI might prevent SUMO1 from modifying key substrates (Luo et al., 2013), although the SUMO1-conjugation pattern as assessed by SDS-PAGE and Western blotting appeared to be very similar in His6 -HA-SUMO1 KI and control samples (Figure 6), excluding major defects […]". The only control in Figure 6 is a SUMO1 KO, it is hard to follow the logic of the argument.*

The corresponding paragraph has been edited for clarity (Results, subsection “Anti-SUMO1 immunolabelling shows no SUMO1 at synapses in His6-HA-SUMO1 KI neurons”).